# TD-MoE: Tensor Decomposition for MoE Models

**Yuebin Xu**[1]     **Yanhong Wang**[1]     **Xuemei Peng**[1]     **Hui Zang**[3†]

**Minghao Chen**[3]     **Pengfei Xia**[3]     **Zeyi Wen**[1,2†]

[1]HKUST (GZ)     [2]HKUST     [3]Huawei

## Abstract

Mixture-of-Experts (MoE) architectures have demonstrated remarkable capabilities and scalability for large language models, but incur a substantial memory footprint due to redundant expert parameters. Existing compression approaches, particularly those based on low-rank decomposition, typically operate at the granularity of individual experts. However, such per-expert methods neglect structural redundancies shared across experts, limiting their compression efficiency and effectiveness. In this work, we introduce TD-MoE (Tensor Decomposition for MoE Compression), a data-aware method that jointly factorizes expert weights by capturing global dependencies. Our contributions are threefold: (i) Cross-expert tensorization with joint three-dimensional decomposition, which unifies all experts within a layer into a single tensor and captures shared structure beyond per-expert scope; (ii) A multi-linear whitening strategy, which decorrelates input and output features, yielding a more balanced and data-adaptive decomposition; (iii) A three-dimensional rank allocation mechanism, which dynamically assigns 3D decomposition ranks across dimensions to best meet a target compression ratio while minimizing the reconstruction error. Extensive experiments on Qwen2-57B-A14B and Mixtral-8×7B across seven commonsense reasoning benchmarks demonstrate that TD-MoE achieves almost lossless performance under 20% parameter reduction, and delivers more than 11% and 14% gains over state-of-the-art decomposition-based baselines at 40% and 60% compression. Further ablation studies validate the effectiveness of each component, highlighting the importance of joint factorization, whitening, and rank allocation.

## 1 Introduction

Mixture-of-Experts (MoE) architectures are key techniques for scaling large language models to trillions of parameters (Cai et al., 2025). By routing tokens to only a subset of experts, they achieve strong scalability, but at the cost of substantial memory overhead from duplicated expert parameters, which severely limits the deployment efficiency in practical large-scale MoE systems (Rajbhandari et al., 2022; Lepikhin et al., 2020).

A natural way to mitigate this overhead is through model compression, and recent research has explored a broad spectrum of strategies, including pruning (Lu et al., 2024), decomposition (Yuan et al., 2023; Wang et al., 2024; Li et al., 2025), merging (Zhou et al., 2025), and quantization (Hu et al., 2025). Pruning-based approaches reduce model size by removing redundant experts or entire MoE layers, while merging and subspace representations fuse functionally similar experts to cut parameter counts. Quantization methods (Huang et al., 2025a) further reduce memory usage by lowering weight precision through adaptive bit-width allocation. Among these, decomposition-based techniques have emerged as a compelling direction. By factorizing the large weight matrices inside experts into low-rank components, decomposition can directly target the densest parameter blocks, yielding substantial compression without altering the model architecture or routing behavior. This property makes decomposition highly scalable and especially appealing for trillion-parameter MoE models, where the majority of memory cost arises from expert weights. A representative example is MoE-SVD (Li et al., 2025), which applies singular value decomposition (SVD) to each expert's weight matrices, effectively reducing the number of parameters while maintaining model accuracy.

---

[†]Corresponding author.

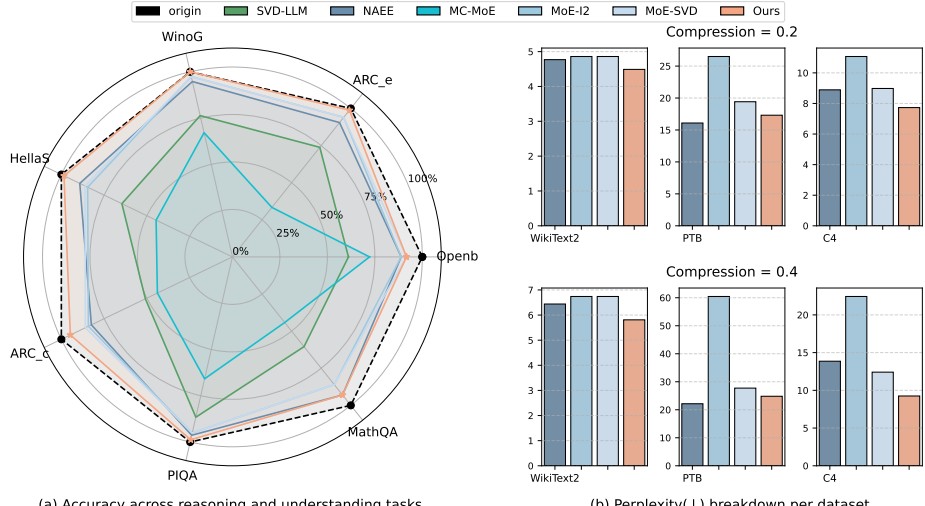

(a) Accuracy across reasoning and understanding tasks.

(b) Perplexity(↓) breakdown per dataset.

Figure 1: **Comprehensive evaluation on Mixtral-8×7B compression.** (a) Accuracy across reasoning and understanding benchmarks at 20% compression ratio. (b) Perplexity (↓) on WikiText-2, PTB, and C4 under 20% and 40% compression. Lower perplexity and higher accuracy indicate better performance.

Empirical studies show that MoE-SVD achieves notable compression ratios with limited perplexity degradation, underscoring the promise of decomposition for large-scale MoE compression. However, most existing decomposition approaches still operate at the level of individual experts (Yang et al., 2024; Li et al., 2025), overlooking the structural redundancies across experts. This per-expert isolation limits their ability to fully leverage shared structure, resulting in suboptimal trade-offs between compression and model performance.

Building on these observations, we contend that leveraging inter-expert structural redundancy is crucial for achieving high-performance MoE compression. While experts within a single layer are typically optimized on related distributions and exhibit correlated weight patterns; yet, current methods like MoE-SVD treat them as isolated entities. This discrepancy motivates a transition from independent expert-wise factorization toward a unified tensor decomposition framework, capable of capturing global correlations across experts.

To this end, we propose TD-MoE (Tensor Decomposition for MoE Compression), a data-aware method that reformulates MoE compression as a joint tensor decomposition problem. Specifically, we stack the weights of all experts in a layer into a three-dimensional tensor and apply Tucker decomposition (Tucker, 1966) to compress them jointly, without altering routing behavior or requiring any post-compression fine-tuning. To enhance the decomposition quality, we introduce a multilinear whitening strategy that decorrelates input and output dimensions before factorization, enabling more balanced and data-adaptive low-rank approximations. In addition, we design a 3D rank allocation mechanism that automatically distributes compression ranks across dimensions, allowing the method to meet a target compression budget. Extensive experiments demonstrate the effectiveness of TD-MoE. On Qwen2-57B-A14B and Mixtral-8×7B, TD-MoE achieves nearly lossless under 20% compression. At more aggressive settings of 40% and 60% compression, TD-MoE delivers over 11% and 14% relative improvement compared to MoE-SVD (Li et al., 2025) and NAEE (Lu et al., 2024) across multiple commonsense reasoning and language modeling tasks. Ablation studies further validate the necessity of each component, highlighting the central role of cross-expert factorization, whitening, and rank allocation. Our contributions are summarized as follows:

- We propose TD-MoE, reformulating MoE compression from per-expert decomposition to a joint 3D tensor decomposition. By stacking all experts in a layer into a three-dimensional tensor, TD-MoE performs cross-expert compression that leverages shared structures and correlations overlooked by expert-wise methods, preserving the original expert routing and architectural semantics.
- We propose a novel multi-linear whitening strategy that decorrelates features across dimensions, thereby balancing the contribution of different modes and enabling joint decomposition to identify

more informative low-rank structures, ultimately achieving better trade-offs between compression budget and accuracy under diverse compression ratios, model architectures, and datasets.

- Experiments demonstrate that TD-MoE delivers superior performance in compressing MoE models. In particular, it delivers nearly lossless accuracy at a 20% compression ratio, and outperforms state-of-the-art decomposition-based baselines by more than 11% under 40–60% compression.

## 2 RELATED WORK

### 2.1 MODEL COMPRESSION FOR MIXTURE-OF-EXPERTS

The large memory footprint of large-scale MoE models has recently brought the research into specialized compression techniques (Cai et al., 2025). These emerging methods primarily fall into three categories: expert pruning, expert merging, and low-rank decomposition, often used in combination. A primary strategy to reduce the MoE parameter count is to decrease the number of experts. This can be achieved by either removing experts entirely via expert pruning or combining similar ones via expert merging. The MC-SMoE framework (Li et al., 2024) exemplifies the merging approach by first clustering experts to identify similarity, then applying a frequency-based merging criterion to combine them. MoE-I2 (Yang et al., 2024) adopts a two-stage hybrid strategy, beginning with inter-expert pruning to discard less critical experts wholesale, followed by compressing the remaining inter-experts by low-rank decomposition. D2-MoE (Gu et al., 2025) first extracts a shared expert, decomposes the computed delta weights via SVD decomposition, and last applies quantization. These methods effectively reduce the expert count based on heuristic criteria, such as similarity, activation frequency, or explicit knowledge sharing. MoE-SVD (Li et al., 2025) enhances per-expert SVD by sharing the input projection matrix across experts and trimming the output projection matrix. This method relies on the strong assumption that expert redundancy is primarily concentrated in a shared input projection space, while functional specialization occurs only in the mapping to the output space. All the above methods impose a complex decision-making process, which requires a decision whether to keep, merge, or discard experts. Inversely, our method offers an alternative solution: instead of removing experts, the tensor decomposition naturally learns a compressed representation in the expert dimension via its expert-mode factor matrix, allowing for a more graceful and data-driven reduction of expert-level redundancy without hard pruning

### 2.2 LOW-RANK AND TENSOR DECOMPOSITION

Low-rank matrix factorization (Golub & Van Loan, 2013), particularly Singular Value Decomposition (SVD), serves as the cornerstone for modern model compression (Yuan et al., 2023; Wang et al., 2024; 2025; Huang et al., 2025b). By identifying and removing small singular values, SVD effectively compresses individual weight matrices in Transformers. Tensor decomposition generalizes this concept to higher-dimensional spaces; mathematically, SVD can be viewed as the second-order instantiation of the Tensor framework. Classical methods of tensor decomposition include Tucker decomposition (Tucker, 1966), CP decomposition (Hitchcock, 1927), and Tensor Train (Oseledets, 2011). These techniques factorize a high-order tensor into a set of factor matrices and, in the case of Tucker, a compact core tensor, thereby reducing storage while retaining expressive capacity. In the deep learning era, tensor decomposition has been explored for compressing CNNs (Phan et al., 2020; Yin et al., 2022; Kim et al., 2015; Zhen et al., 2022) and RNNs (Novikov et al., 2015). These architectures are naturally suitable to high-order factorization because components like convolutional kernels possess inherent multi-dimensional structures (e.g., spatial height, width, and channels). Despite these advances, most existing applications focus on individual weight matrices, applying tensorization by reshaping a single matrix into a higher-order tensor before decomposition. However, it does not directly translate to transformer-based MoE models, where experts are independent weight matrices without an inherent higher-order organization. Our work departs from this conventional perspective by introducing a cross-expert tensorization that explicitly stacks expert weights into a three-dimensional tensor, thereby exposing inter-expert redundancy for compression.

# 3 METHOD

We introduce TD-MoE (Tensor Decomposition for MoE), a novel method for MoE compression through joint tensor decomposition. It consists of three components: cross-expert tensorization, multi-linear whitening, and 3D rank allocation. The pipeline operates as follows: given the weight matrices of all experts in a layer, we first stack them into a three-dimensional tensor that captures both expert diversity and cross-expert correlations. Next, we apply a whitening transformation along the input or output modes using activation statistics, which decorrelates feature dimensions and makes the subsequent decomposition more balanced and data-adaptive. We then perform Tucker decomposition on the whitened tensor to obtain a compact core tensor and factor matrices, with their sizes determined by an adaptive 3D rank allocation scheme that respects a global compression budget. Finally, the decomposed factors are re-colored via the inverse whitening matrices to reconstruct the compressed expert weights. The main framework is illustrated in Figure 2.

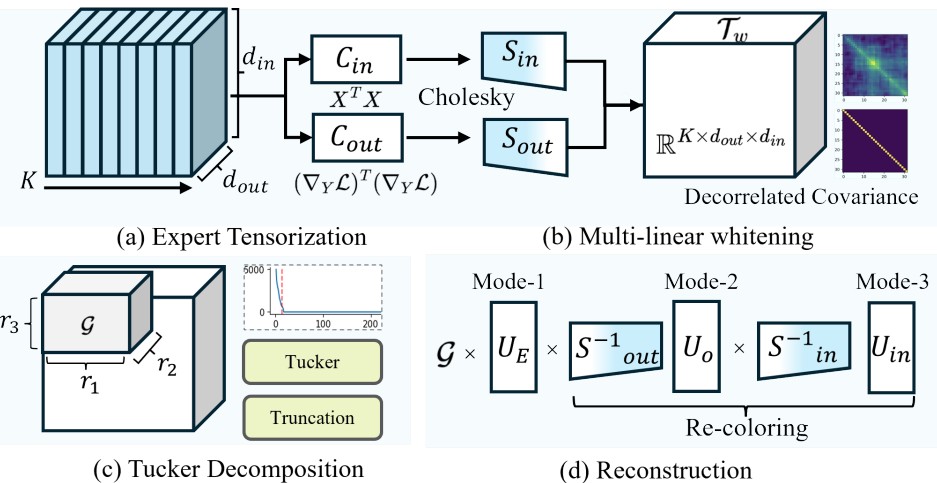

Figure 2: **Overall framework of TD-MoE.** (a) Expert Tensorization: expert weights are stacked into a 3D tensor $\mathcal{T}$; (b) Multi-linear Whitening: input/output modes are whitened to remove cross-dimensional correlations, producing $\mathcal{T}_w$; (c) Tucker Decomposition: $\mathcal{T}_w$ is factorized into a low-rank core $\mathcal{G}$ with ranks $(r_1, r_2, r_3)$; (d) Reconstruction: factor matrices $U_o$ and $U_{in}$ are re-colored via inverse whitening to obtain the final low-rank approximation.

## 3.1 PROBLEM FORMULATION

We formalize the compression objective for an MoE layer with $K$ experts, where the $i$-th expert is parameterized by $W^{(i)} \in \mathbb{R}^{d_{out} \times d_{in}}$. Given a calibration distribution $\mathcal{D}_{\text{calib}}$, the goal is to find compressed weights $\{\hat{W}^{(i)}\}$ that preserve the activation behavior of the original model:

$$\min_{\{\hat{W}^{(i)}\}} \mathbb{E}_{x \sim \mathcal{D}_{\text{calib}}} \left[ \sum_{i=1}^{K} \|W^{(i)}x - \hat{W}^{(i)}x\|_2^2 \right]. \qquad (1)$$

Existing methods, such as MoE-SVD (Li et al., 2025), apply SVD independently to each $W^{(i)}$, truncating its factors to fit a parameter budget. This per-expert isolation overlooks redundancies across experts, thus limiting compression efficiency and leading to sharp performance drops at higher compression ratios.

## 3.2 JOINT TUCKER FACTORIZATION

**Cross-Expert Tensorization.** Instead of compressing experts individually, we adopt a global, ensemble-level view. We stack the $K$ expert matrices into a three-dimensional tensor $\mathcal{T}$:

$$\mathcal{T} \in \mathbb{R}^{K \times d_{out} \times d_{in}},$$

where mode-1 indexes experts, mode-2 represents output features, and mode-3 represents input features. This tensorization unifies the ensemble into a single object, enabling joint modeling of intra-expert structure and inter-expert redundancies. Unlike pruning or merging, which make hard expert-level decisions, this formulation provides a principled basis for multi-linear factorization and data-aware compression.

**Tucker Decomposition.** Tucker decomposition generalizes principal component analysis to higher-order tensors. Given $\mathcal{T} \in \mathbb{R}^{d_1 \times d_2 \times d_3}$, it factorizes the tensor into a compact core $\mathcal{G} \in \mathbb{R}^{r_1 \times r_2 \times r_3}$ and factor matrices $\mathbf{U}^{(n)} \in \mathbb{R}^{d_n \times r_n}$ for $n = 1, 2, 3$:

$$\mathcal{T}_w \approx \mathcal{G} \times_1 \mathbf{U}^{(1)} \times_2 \mathbf{U}^{(2)} \times_3 \mathbf{U}^{(3)} \tag{2}$$

where the ranks $(r_1, r_2, r_3)$ control both compression and reconstruction fidelity[*]. For MoE compression, we jointly model all experts as a three-dimensional tensor $\mathcal{T} \in \mathbb{R}^{K \times d_{out} \times d_{in}}$, where $K$ is the number of experts. Instead of decomposing each expert in isolation, we apply Tucker decomposition directly to the whitened tensor $\mathcal{T}_w$, yielding:

$$\min_{\mathcal{G}, U_1, U_2, U_3} \|\mathcal{T}_w - \mathcal{G} \times_1 U_1 \times_2 U_2 \times_3 U_3\|_F^2, \tag{3}$$

where $U_1 \in \mathbb{R}^{K \times r_1}$, $U_2 \in \mathbb{R}^{d_{out} \times r_2}$, and $U_3 \in \mathbb{R}^{d_{in} \times r_3}$. The factors admit natural interpretations in the MoE setting: $U_1$ forms $r_1$ meta-experts compressing inter-expert redundancy, while $U_2$ and $U_3$ define low-dimensional output and input subspaces. A Tucker-compressed MoE layer contains $r_1 r_2 r_3$ core parameters and $(K r_1 + d_{out} r_2 + d_{in} r_3)$ factor-matrix parameters, yielding $P_{\text{tucker}} = r_1 r_2 r_3 + (K r_1 + d_{out} r_2 + d_{in} r_3)$, compared to the original $P_{\text{orig}} = K d_{out} d_{in}$. This closed-form expression provides direct control over the desired compression ratio (Section 3.4). Overall, Tucker decomposition offers a unified mechanism that simultaneously compresses feature dimensions and removes cross-expert redundancy in a principled manner.

## 3.3 MULTI-LINEAR TENSOR WHITENING

Directly decomposing raw expert weights can be suboptimal because input activations are often highly correlated, making the feature space ill-conditioned (Yuan et al., 2023). As a result, the singular values of each $W^{(i)}$ do not faithfully reflect their contribution to the model's behavior, leading to inferior compression decisions. Prior work addresses this issue by applying whitening transformations that decorrelate features and rescale them to unit variance before decomposition (Wang et al., 2024; Li et al., 2025).

**Multi-Linear Whitening.** We extend the whitening to the tensorized MoE setting. Given a calibration dataset $\mathcal{D}_{\text{calib}}$, we collect both input activations $X \in \mathbb{R}^{N \times d_{in}}$ and output gradients $\nabla_Y \mathcal{L} \in \mathbb{R}^{N \times d_{out}}$, where $N$ is the number of tokens. These statistics define the input and output covariances, $\Sigma_{in} = \frac{1}{N} X^T X$ and $\Sigma_{out} = \frac{1}{N} (\nabla_Y \mathcal{L})^T (\nabla_Y \mathcal{L})$. Here, $\Sigma_{in}$ captures correlations among input features, while $\Sigma_{out}$ reflects output sensitivities with respect to the loss. To obtain whitening transformations, we compute regularized square-root inverses of these covariances, namely $S_{in} = (\Sigma_{in} + \epsilon I)^{-\frac{1}{2}}$ and $S_{out} = (\Sigma_{out} + \epsilon I)^{-\frac{1}{2}}$, where $\epsilon$ is a small constant for numerical stability. Applying these to expert tensor $\mathcal{T}$ yields the whitened representation:

$$\mathcal{T}_w = \mathcal{T} \times_2 S_{out} \times_3 S_{in}. \tag{4}$$

This *multi-linear whitening* can decorrelate input or output modes, or both simultaneously, producing a well-conditioned tensor that improves the stability and effectiveness of Tucker decomposition. Unlike prior 2-D whitening approaches, our tensor-level whitening leverages multi-linear structure and explicitly adapts to input or output statistics, ensuring that TD-MoE not only captures cross-expert redundancy but also aligns the decomposition with the statistical geometry of data.

---

[*]The mode-$n$ product is $(\mathcal{X} \times_n U)_{i_1 \dots j \dots i_3} = \sum_k \mathcal{X}_{i_1 \dots k \dots i_3} U_{kj}$, and the equivalent element-wise reconstruction is $\mathcal{T}_{w, i_1 i_2 i_3} = \sum_{r_1, r_2, r_3} \mathcal{G}_{r_1 r_2 r_3} U^{(1)}_{i_1 r_1} U^{(2)}_{i_2 r_2} U^{(3)}_{i_3 r_3}$.

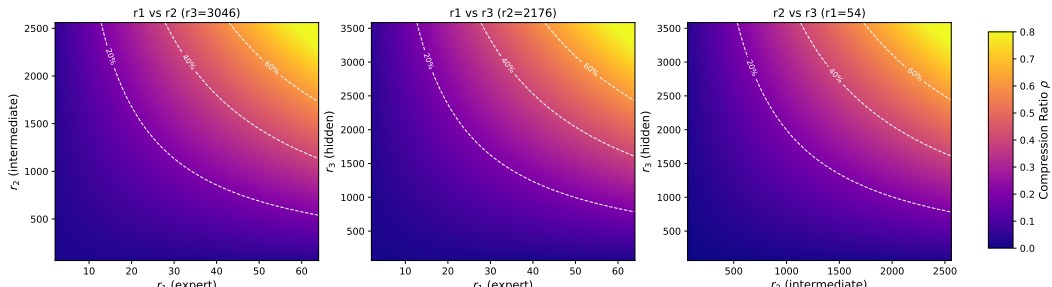

Figure 3: **Parameter compression landscape on Qwen2-57B-A14B.** Each subplot varies two Tucker ranks while fixing the third at a high-capacity value ($K = 64, d_{\text{int}} = 2560, d_{\text{hid}} = 3584$): (a) $r_1$ vs. $r_2$; (b) $r_1$ vs. $r_3$; (c) $r_2$ vs. $r_3$. White dashed contours highlight $\rho \in \{20\%, 40\%, 60\%\}$.

**Re-coloring for Inference.** To deploy compressed models efficiently, we absorb the inverse whitening transforms into the Tucker factors. Specifically, the original expert tensor is recovered as $\mathcal{T} \approx \mathcal{G} \times_1 U_1 \times_2 (S_{out}^{-1} U_2) \times_3 (S_{in}^{-1} U_3)$. For inference, we store the pre-colored factors $U'^{(1)} = U_1$, $U'^{(2)} = S_{out}^{-1} U_2$, and $U'^{(3)} = S_{in}^{-1} U_3$, ensuring that whitening and re-coloring introduce no extra runtime cost. We next address how to adaptively allocate Tucker ranks $(r_1, r_2, r_3)$ to balance compression efficiency and model fidelity.

### 3.4 ADAPTIVE 3D RANK ALLOCATION

We consider an MoE weight tensor $\mathcal{T} \in \mathbb{R}^{K \times d_{out} \times d_{in}}$ with original parameter count $P_{\text{orig}}$ and Tucker-decomposed size $P_{\text{tucker}}$. To satisfy a target compression ratio $\rho^*$, we introduce an adaptive three-dimensional rank allocation scheme that searches over the Tucker rank triplet $(r_1, r_2, r_3)$ under the constraint $P_{\text{tucker}} \approx (1 - \rho^*) P_{\text{orig}}$. For candidate pair $(r_1, r_2)$, the feasible $r_3$ is obtained in closed form from the budget equation:

$$r_3 = \frac{(1 - \rho^*) P_{\text{orig}} - (K r_1 + d_{out} r_2)}{r_1 r_2 + d_{in}}, \tag{5}$$

followed by projection onto the valid range $1 \le r_3 \le d_{in}$. This formulation reduces the original 3D search to a two-dimensional sweep over $(r_1, r_2)$ with a one-dimensional closed-form update for $r_3$, enabling efficient exploration of the rank space under strict parameter budgets. We select the rank triplet that minimizes deviation from the target model size subject to integer and box constraints. By sweeping $r_1 \in [1, K]$ and $r_2 \in [1, d_{out}]$ and computing the corresponding $r_3$ from Eq. 5, this method identifies decompositions that tightly match the desired compression ratio $\rho^*$. Figure 3 visualizes the induced compression landscape, where each subplot varies two ranks while fixing the third. The colormap encodes the realized compression ratio, and contour lines mark target levels (20%, 40%, 60%), revealing how feasible configurations form structured regions in the rank space and showing how it systematically selects solutions lying on or near the desired contours, ensuring both feasibility and compression budget. Algorithmic details appear in Appendix C.

## 4 EXPERIMENTS

### 4.1 EXPERIMENTAL SETUP

**Models and Tasks.** We evaluate TD-MoE on modern MoE models with varying numbers of experts: Qwen2-57B-A14B (Team et al., 2024) with 64 standard and 8 shared experts, Mixtral-8x7B (Jiang et al., 2024) with 8 experts, and Phi-3.5-MoE (Abdin et al., 2024) with 16 experts. Post-compression performance is assessed on a diverse set of tasks to ensure a comprehensive evaluation. We evaluate our method on 10 tasks, including commonsense reasoning and language modeling. For commonsense reasoning, we evaluate zero-shot accuracy on seven benchmarks: OpenbookQA (Mihaylov et al., 2018), WinoGrande (Sakaguchi et al., 2021), HellaSwag (Zellers et al., 2019), PIQA (Bisk et al., 2020), MathQA (Amini et al., 2019), ARC-easy, and ARC-challenge (Clark et al., 2018). All zero-shot evaluations are conducted using the LM-Evaluation-Harness framework (Gao et al.,

2021) to ensure reproducibility and fair comparison with prior work. For language modeling, we measure Perplexity (PPL) on WikiText-2 (Merity et al., 2017), Penn Treebank (PTB) (Marcus & Marcinkiewicz, 1994), and a held-out test set from C4 (Raffel et al., 2020).

**Implementation Details.** All experiments use a fixed calibration set of 256 WikiText-2 samples to compute whitening statistics, and all results are obtained under a strict post-training setup *without fine-tuning*. Tucker decomposition is implemented in PyTorch (Paszke et al., 2019) and TensorLy (Kossaifi et al., 2019), and covariance eigenvalues below $10^{-3}$ are clipped for stability. We evaluate our method in two whitening versions under two complementary truncation settings. Full 3-mode decomposition and truncation are applied on Qwen2-57B-A14B to compress its 64-expert layers, validating the efficacy in large-scale expert configurations. An expert-preserving scheme is applied on Mixtral-8×7B to ensure a fair comparison with MoE-SVD (Li et al., 2025), whose formulation keeps the expert dimension fixed. Pruning and quantization are orthogonal to tensor decomposition; we thus evaluate their synergistic effects separately in Section 4.3 to demonstrate the broad compatibility of TD-MoE with other compression paradigms in practical pipelines.

## 4.2 COMPRESSION PERFORMANCE COMPARISON

**Commonsense Reasoning Performance.** The right portion of Table 1 reports accuracy on seven commonsense reasoning tasks across multiple compression ratios validated on two MoE models. Three consistent trends emerge. (a) At the 20% compression level, TD-MoE preserves accuracy remarkably well on both models, with less than 1% absolute drop relative to the original model. Both input- and output-whitening variants outperform strong baselines, and the output-whitening variant achieves improvements of up to four points (4% and 7% relative gains). Notably, the gains are most pronounced on reasoning-intensive tasks such as HellaSwag and ARC-c, which are highly sensitive to expert representation quality. (b) At the 40% compression level, TD-MoE remains further robust. The output-whitening variant reaches 0.56 on Qwen2-57B-A14B and 0.57 on Mixtral-8×7B, yielding 17% and 14% relative improvements over MoE-SVD. The input-whitening variant shows similarly stable behavior, demonstrating the regularization and conditioning benefits of whitening during Tucker truncation. (c) At the 60% compression level, performance degradation is unavoidable across all methods due to the aggressive parameter reduction. However, TD-MoE remains substantially more resilient: on both models it delivers large margins of 11% and 21.6% relative improvement over MoE-SVD. These results indicate that whitening-based Tucker decomposition preserves structural information more effectively than these baselines, especially under high compression. Additional broad experimental results on Phi-3.5-MoE are provided in Appendix E.1.

**Language Modeling Perplexity Comparison.** The left portion of Table 1 reports perplexity on WikiText-2, PTB, and C4 for Qwen2-57B-A14B and Mixtral-8×7B across multiple compression ratios. At 20% compression, the output-whitening variant demonstrates strong performance (e.g., 4.49/17.20/7.73 on Mixtral-8×7B), clearly outperforming NAEE and MoE-SVD, while the input-whitening variant remains competitive. At 40% compression, baselines degrade sharply, whereas TD-MoE maintains robust low perplexity (5.79/24.60/9.21), highlighting clear advantages. At 60% compression, perplexity rises for all methods, but TD-MoE degrades more gracefully. We also observe that output whitening performs slightly better at lower compression, while input whitening becomes more stable under more aggressive truncation. This observation aligns with the role of whitening in mitigating error propagation. At lower compression levels, output whitening more effectively preserves the dominant directions of the output distribution. Conversely, under higher compression, input whitening stabilizes the more aggressive rank-reduction by preventing amplification of poorly conditioned input activations. These results demonstrate that our Tucker decomposition method consistently outperforms strong baselines across varying compression scales and models.

## 4.3 ABLATION STUDIES

**Whitening Analysis.** We analyze how multi-linear whitening affects the numerical structure of activations and the resulting Tucker decomposition. Beyond end-to-end performance comparison in Table 1, we quantify whitening's impact on covariance spectra and cross-dimensional correlations as reported in Table 2. Observations include: (a) Before whitening, activation covariances exhibit extremely ill-conditioned spectra, with eigenvalues spanning nearly four orders of magnitude (e.g., 9 to $5.4 \times 10^4$ in Layer 9). After whitening, all eigenvalues collapse to 1.0 with maximum deviations

Table 1: Perplexity and accuracy under budget–constrained compression. Top: Qwen2-57B-A14B. Bottom: Mixtral-8×7B. Lower perplexity and higher accuracy indicate better performance.

| Ratio | Method | Wiki. | PTB | C4 | Openb. | ARC_e | WinoG. | HellaS. | ARC_c | PIQA | MathQA | Avg. |
|---|---|---|---|---|---|---|---|---|---|---|---|---|
| | **Qwen2-57B-A14B** (8+64 experts, 8+top-8 activated) | | | | | | | | | | | |
| 0% | Original | 5.12 | 9.18 | 8.86 | 0.33 | 0.75 | 0.74 | 0.63 | 0.46 | 0.81 | 0.39 | 0.59 |
| 20% | NAEE (2024) | 6.96 | 10.39 | 10.52 | 0.31 | 0.72 | 0.72 | 0.59 | 0.42 | 0.79 | 0.36 | 0.56 |
| | MoE-SVD (2025) | **5.41** | 13.26 | 11.63 | 0.30 | 0.73 | 0.73 | **0.61** | 0.45 | 0.78 | 0.35 | 0.56 |
| | TD-MoE (input) | 6.65 | 10.40 | **10.34** | 0.30 | **0.79** | **0.73** | 0.58 | **0.49** | 0.80 | **0.37** | **0.58**(↑4%) |
| | TD-MoE (output) | 6.67 | **10.28** | 10.35 | **0.31** | **0.79** | **0.73** | 0.59 | 0.49 | 0.80 | 0.37 | **0.58**(↑4%) |
| 40% | NAEE (2024) | 8.79 | 12.64 | 13.26 | 0.28 | 0.72 | 0.71 | 0.54 | 0.39 | 0.76 | 0.32 | 0.53 |
| | MoE-SVD (2025) | 13.43 | 23.88 | 18.83 | 0.29 | 0.63 | 0.65 | 0.45 | 0.33 | 0.71 | 0.31 | 0.48 |
| | TD-MoE (input) | **7.96** | **12.37** | **12.91** | **0.31** | **0.77** | **0.71** | 0.54 | 0.44 | 0.78 | **0.36** | **0.56**(↑6%) |
| | TD-MoE (output) | 8.07 | 12.50 | 13.09 | 0.29 | **0.77** | **0.71** | 0.54 | 0.45 | 0.78 | 0.35 | **0.56**(↑6%) |
| 60% | NAEE (2024) | 14.79 | 23.52 | 25.00 | 0.21 | 0.58 | 0.61 | 0.44 | 0.29 | 0.69 | 0.26 | 0.44 |
| | MoE-SVD (2025) | 19.46 | 29.94 | 25.06 | 0.27 | 0.62 | 0.64 | 0.44 | 0.32 | 0.69 | 0.30 | 0.47 |
| | TD-MoE (input) | **12.21** | **19.87** | **20.49** | **0.28** | **0.73** | **0.69** | 0.45 | **0.41** | **0.75** | **0.31** | **0.52**(↑11%) |
| | TD-MoE (output) | 12.49 | 19.92 | 21.04 | **0.28** | **0.73** | 0.68 | **0.46** | **0.41** | 0.74 | **0.31** | **0.51**(↑9%) |
| | **Mixtral-8×7B** (8 experts, top-2 activated) | | | | | | | | | | | |
| 0% | Original | 3.84 | 14.70 | 7.18 | 0.35 | 0.84 | 0.76 | 0.65 | 0.57 | 0.82 | 0.43 | 0.63 |
| 20% | NAEE (2024) | 4.77 | 16.09 | 8.89 | 0.32 | 0.76 | 0.72 | 0.58 | 0.47 | 0.79 | 0.40 | 0.58 |
| | MoE-SVD (2025) | 4.86 | 19.42 | 8.98 | 0.33 | 0.79 | 0.74 | 0.56 | 0.49 | 0.78 | 0.37 | 0.58 |
| | TD-MoE (input) | 4.67 | 19.82 | 8.04 | 0.32 | 0.82 | 0.76 | 0.61 | 0.53 | 0.82 | 0.40 | 0.61 (↑5%) |
| | TD-MoE (output) | **4.49** | 17.20 | **7.73** | **0.33** | **0.83** | **0.77** | **0.64** | 0.53 | **0.82** | 0.40 | **0.62**(↑7%) |
| 40% | NAEE (2024) | 6.44 | 22.15 | 13.86 | 0.25 | 0.63 | 0.64 | 0.46 | 0.36 | 0.72 | 0.35 | 0.48 |
| | MoE-SVD (2025) | 6.74 | 27.73 | 12.41 | 0.27 | 0.72 | 0.67 | 0.43 | 0.38 | 0.71 | 0.32 | 0.50 |
| | TD-MoE (input) | 7.17 | 32.10 | 11.44 | 0.28 | 0.76 | 0.72 | 0.49 | 0.43 | 0.78 | 0.34 | 0.54 (↑8%) |
| | TD-MoE (output) | **5.79** | 24.60 | **9.21** | **0.28** | **0.77** | **0.76** | **0.57** | **0.47** | **0.79** | 0.35 | **0.57**(↑14%) |
| 60% | NAEE (2024) | 11.43 | 47.28 | 31.16 | 0.17 | 0.42 | 0.55 | 0.33 | 0.23 | 0.62 | 0.26 | 0.36 |
| | MoE-SVD (2025) | 13.52 | 130.26 | 39.54 | 0.19 | 0.45 | 0.55 | 0.33 | 0.23 | 0.62 | 0.25 | 0.37 |
| | TD-MoE (input) | 13.64 | **66.00** | **19.63** | **0.22** | **0.58** | **0.63** | **0.40** | **0.34** | **0.73** | **0.27** | **0.45** (↑22%) |
| | TD-MoE (output) | 17.78 | 79.43 | 24.85 | 0.21 | 0.55 | 0.62 | 0.38 | 0.28 | 0.65 | 0.24 | **0.42** (↑14%) |

below $10^{-7}$ (rightmost column in Table 2), removing scale anisotropy and stabilizing truncated Tucker factors. (b) Off-diagonal correlations reach 0.63–0.79 pre-whitening with standard deviations around $10^{-2}$; whitening eliminates these entirely, reducing all residual correlations to below $10^{-7}$ and producing fully decorrelated activation subspaces. These numerical improvements translate into consistent empirical gains across datasets, with both input and output whitening yielding noticeably more stable performance under compression.

Table 2: Covariance spectra and cross-dimensional correlations before and after whitening.

| Layer | $\lambda_{\min}-\lambda_{\max}$ (Pre) | Corr Std / Max (Pre) | Post-Whitening Deviation |
|---|---|---|---|
| 3 | $22 - 3.9 \times 10^4$ | $1.06 \times 10^{-2}$ / 0.72 | $1.0 \times 10^{-7}$ |
| 5 | $31 - 3.8 \times 10^4$ | $1.14 \times 10^{-2}$ / 0.64 | $3.0 \times 10^{-8}$ |
| 7 | $18 - 4.3 \times 10^4$ | $1.27 \times 10^{-2}$ / 0.78 | $6.0 \times 10^{-8}$ |
| 9 | $9 - 5.4 \times 10^4$ | $1.42 \times 10^{-2}$ / 0.63 | $1.1 \times 10^{-7}$ |
| 12 | $10 - 5.3 \times 10^4$ | $1.34 \times 10^{-2}$ / 0.76 | $1.0 \times 10^{-7}$ |
| 24 | $24 - 1.9 \times 10^4$ | $1.01 \times 10^{-2}$ / 0.79 | $4.0 \times 10^{-8}$ |

**Complexity and Runtime Analysis.** We evaluate the computational and memory overhead of the whitening process, Cholesky decomposition, and Tucker decomposition within our method on NVIDIA A800 GPUs, with results summarized in Table 3. Specifically, the whitening process introduces negligible overhead, consuming less than 1% additional memory and 0.11 TFLOPs, and its cost remains invariant to the expert count since it is applied to a 2D covariance matrix once per layer. Then, Cholesky decomposition scales robustly to large FFN dimensions (e.g., 14,336) and does not constitute a practical bottleneck. For the core decomposition, our randomized Tucker implementation exhibits a superior or comparable

Table 3: Overhead and runtime.

| Component | Mixtral-8×7B | Phi-3.5-MoE |
|---|---|---|
| Whitening Mem. | 890 MB | 231 MB |
| Whitening FLOPs | 0.11 TFLOPs | 0.02 TFLOPs |
| Cholesky (max dims) | 784 MB | 156 MB |
| Tucker (rand) / SVD | 20.7 s / 9.4 s | 7.1 s / 17.7 s |
| Relative Speed | 1.3–2.2× slower | 2.5–4.6× faster |

complexity of $O(d_{out}d_{in}r)$ per step, contrasting with the $E \cdot O(d_{out}d_{in}\min(d_{out}, d_{in}))$ required by per-expert SVD. This efficiency is particularly evident on models with higher expert counts. This suggests that TD-MoE is highly scalable for modern MoE architectures, where the large number of experts would make independent SVD expensive. By decoupling the decomposition cost from the expert dimension, our method provides a principled framework for compressing increasingly wide MoE layers. Importantly, these computations are conducted offline, incurring no inference overhead.

**Compatibility Analysis.** We evaluate the extensibility of TD-MoE by integrating it with common post-training compression techniques on Mixtral-8x7B. Specifically, we examine: (i) NF4 quantization applied post-decomposition, and (ii) structured pruning of low-energy core slices and their corresponding factor-matrix columns. As reported in Table 4, the quantization brings marginal performance improvements with 0.1 PPL in LM Avg, while reasoning accuracy (OpenbookQA, Arc-E, Wino, Arc-C, PIQA) remains virtually unchanged. Similarly, structured pruning follows a smooth, monotonic degradation pattern; for instance, at 20% compression, increasing sparsity to 40% results in a graceful decline of Reason Avg from 0.66 to 0.61. These results confirm that TD-MoE is highly compatible with other compression

Table 4: Synergy of TD-MoE with NF4 quantization and structured pruning.

| Setting | Prune | LM Avg | Rea. Avg |
|---------|-------|--------|----------|
| Original | – | 8.57 | 0.63 |
| *20% Compression* | | | |
| TD-MoE | 0% | 9.81 | 0.66 |
| + NF4 | 0% | 9.70 | 0.66 |
| + Pruning | 20% | 12.72 | 0.64 |
| + Pruning | 30% | 14.21 | 0.62 |
| + Pruning | 40% | 15.72 | 0.61 |
| *40% Compression* | | | |
| TD-MoE | 0% | 13.20 | 0.57 |
| + NF4 | 0% | 13.19 | 0.56 |
| + Pruning | 40% | 30.06 | 0.54 |

paradigms. Specifically, our pruning strategy operates within the Tucker domain by ranking and removing core slices based on their energy. This demonstrates that the Tucker-decomposed structure is highly amenable to structured pruning. Implementation details are provided in Appendix D.

**Hyperparameter Sensitivity Analysis.** We investigate the robustness of TD-MoE with respect to two key whitening-related hyperparameters: the calibration set size $N$ and the eigenvalue clipping threshold $\tau$. As shown in Table 5, increasing $N$ from 128 to 2K (including a corpus shift from WikiText-2 to PTB) results in marginal fluctuations with perplexity shifting by less than 0.03 and the downstream accuracy (ARC-E, WinoG, ARC-C, PIQA) by less than 0.01. Similarly, sweeping $\tau$ across four orders of magnitude ($10^{-1}$–$10^{-4}$) yields stable performance under both 20% and 40% compression, with variations within 0.1 PPL and 0.01 accuracy. Notably, the relative ranking of compression settings remains unchanged across all configurations, suggesting that whitening behaves in a well-conditioned regime. These findings demonstrate that TD-MoE is numerically robust and insensitive to the choice of calibration data and clipping threshold

Table 5: Sensitivity of TD-MoE to calibration size $N$ and clipping threshold $\tau$.

| Setting | Avg PPL | Avg Acc | Variation |
|---------|---------|---------|-----------|
| $N = \{128, 256, 512, 1k, 2k\}$ | 9.81–9.84 | 0.73–0.74 | $\Delta$PPL $\leq 0.03$, $\Delta$Acc $\leq 0.01$ |
| $\tau \in \{10^{-1}, 10^{-2}, 10^{-3}, 10^{-4}\}$ (20%) | 7.23–7.32 | 0.73–0.74 | $\Delta$PPL $\leq 0.09$, $\Delta$Acc $\leq 0.01$ |
| $\tau \in \{10^{-1}, 10^{-2}, 10^{-3}, 10^{-4}\}$ (40%) | 10.13–10.20 | 0.69–0.70 | $\Delta$PPL $\leq 0.07$, $\Delta$Acc $\leq 0.01$ |

## 5 CONCLUSION

In this paper, we introduced TD-MoE, a cross-expert tensor decomposition method that rethinks MoE compression from a global perspective. By unifying expert weights into a 3D tensor, incorporating robust multi-linear whitening, and allocating ranks through principled budget-aware strategies, our method offers a scalable method for reducing cross-and intra-expert redundancy while preserving model fidelity. Extensive experiments demonstrate that TD-MoE achieves competitive or superior accuracy under substantial compression ratios and is well-compatible with post-training quantization and pruning. Beyond demonstrating its effectiveness on MoE models, this work highlights the broader potential of the tensor-based compression method as a foundation for more efficient, adaptable, and widely deployable large-scale language models.

## ACKNOWLEDGMENTS

This work is supported by National Key R&D Program of China under Grant No. 2024YFA1012700. It is also funded by the NSFC Project (No. 62306256) and the Natural Science Foundation of Guangdong Province (No. 2025A1515010261).

## ETHICS STATEMENT

This work adheres to the ICLR Code of Ethics. This work does not raise any specific ethical concerns. It focuses on model compression techniques for Mixture-of-Experts models and does not involve sensitive data, human subjects, or high-risk applications. All datasets used, including OpenbookQA, WinoGrande, HellaSwag, PIQA, MathQA, ARC-easy, ARC-challenge, WikiText-2, PTB, and C4 are sourced in compliance with relevant usage guidelines, ensuring no violation of privacy.

## REPRODUCIBILITY STATEMENT

Section 4 and the Appendix document the implementation pipeline and include algorithmic descriptions of the proposed method. Model configurations, compression settings, and evaluation protocols are reported, together with calibration selection and software environments.

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

## A  USE OF LLMS

LLMs were used for auxiliary support, including literature search assistance, grammatical editing, and code debugging. Their use does not constitute intellectual contributions to this work.

## B  DETAILS OF THE METHOD

We provide the detailed procedure of TD-MoE in Algorithm 1. The algorithm takes the original expert weights, a calibration dataset, and a target compression ratio as inputs, and returns a compact Tucker core together with re-colored factor matrices for inference. It first stacks all experts into a unified three-dimensional tensor, applies multi-linear whitening using calibration statistics, and then performs budget-aware Tucker decomposition. Finally, the inverse whitening transformations are absorbed into the factor matrices, ensuring that the compressed representation introduces no additional inference-time computation.

---

**Algorithm 1** Tensor Decomposition for Mixture-of-Experts (TD-MoE)

---

**Require:** Original expert weights $\{W^{(i)} \in \mathbb{R}^{d_{out} \times d_{in}}\}_{i=1}^{K}$; Calibration dataset $\mathcal{D}_{\text{calib}}$; Target compression ratio $\rho$; Regularization constant $\epsilon$.
**Ensure:** Compressed core tensor $\mathcal{G} \in \mathbb{R}^{r_1 \times r_2 \times r_3}$; Re-colored factor matrices $U'^{(1)}, U'^{(2)}, U'^{(3)}$.

    **// Step 1: Cross-Expert Tensorization**
1: Initialize a tensor $\mathcal{T} \in \mathbb{R}^{K \times d_{out} \times d_{in}}$.
2: **for** $i = 1$ to $K$ **do**
3:     $\mathcal{T}[i, :, :] \leftarrow W^{(i)}$         ▷ Stack expert weights into a 3D tensor
4: **end for**

    **// Step 2: Collect Statistics for Whitening**
5: $X, \nabla_Y \mathcal{L} \leftarrow \text{CollectActivationsAndGradients}(\{W^{(i)}\}, \mathcal{D}_{\text{calib}})$
6: $\Sigma_{in} \leftarrow \frac{1}{N} X^T X$         ▷ Compute input covariance
7: $\Sigma_{out} \leftarrow \frac{1}{N} (\nabla_Y \mathcal{L})^T (\nabla_Y \mathcal{L})$         ▷ Compute output covariance

    **// Step 3: Multi-Linear Tensor Whitening**
8: $S_{in} \leftarrow (\Sigma_{in} + \epsilon I)^{-\frac{1}{2}}$         ▷ Compute input whitening matrix via Cholesky
9: $S_{out} \leftarrow (\Sigma_{out} + \epsilon I)^{-\frac{1}{2}}$         ▷ Compute output whitening matrix
10: $\mathcal{T}_w \leftarrow \mathcal{T} \times_2 S_{out} \times_3 S_{in}$         ▷ Apply whitening to input/output modes

    **// Step 4: Adaptive 3D Rank Allocation**
11: $(r_1, r_2, r_3) \leftarrow \text{FindOptimalRanks}(\rho, K, d_{out}, d_{in})$     ▷ Search for ranks satisfying the budget

    **// Step 5: Joint Tucker Factorization**
12: $\mathcal{G}, \{U_1, U_2, U_3\} \leftarrow \text{TuckerDecomposition}(\mathcal{T}_w, \text{ranks} = (r_1, r_2, r_3))$

    **// Step 6: Re-coloring for Inference**
13: $S_{in}^{-1} \leftarrow (\Sigma_{in} + \epsilon I)^{\frac{1}{2}}$
14: $S_{out}^{-1} \leftarrow (\Sigma_{out} + \epsilon I)^{\frac{1}{2}}$
15: $U'^{(1)} \leftarrow U_1$
16: $U'^{(2)} \leftarrow S_{out}^{-1} U_2$         ▷ Absorb inverse whitening into output factor ($d_{out}$)
17: $U'^{(3)} \leftarrow S_{in}^{-1} U_3$         ▷ Absorb inverse whitening into input factor ($d_{in}$)
18: **return** $\mathcal{G}, U'^{(1)}, U'^{(2)}, U'^{(3)}$

---

## C  RANK ALLOCATION ALGORITHM

The goal of rank allocation is to find Tucker ranks $(r_1, r_2, r_3)$ that match the target parameter budget as closely as possible. For an MoE expert tensor of size $K \times d_{\text{out}} \times d_{\text{in}}$, the original and Tucker-compressed parameter counts are

$$P_{\text{orig}} = K d_{\text{out}} d_{\text{in}},$$

and
$$P_{\text{tucker}}(r_1, r_2, r_3) = r_1 r_2 r_3 + K r_1 + d_{\text{out}} r_2 + d_{\text{in}} r_3.$$
Given a target compression ratio $\rho^*$, the target parameter budget is
$$P_{\text{target}} = (1 - \rho^*) P_{\text{orig}}.$$
For each candidate pair $(r_1, r_2)$, we solve the budget equation
$$P_{\text{tucker}}(r_1, r_2, r_3) = P_{\text{target}}$$
for the corresponding feasible value of $r_3$:
$$r_3 = \frac{P_{\text{target}} - (K r_1 + d_{\text{out}} r_2)}{r_1 r_2 + d_{\text{in}}}.$$
We then iterate over $r_1 \in [1, K]$ and $r_2 \in [1, d_{\text{out}}]$, compute the corresponding integer candidates for $r_3$, and retain only feasible values satisfying $1 \le r_3 \le d_{\text{in}}$. Among all feasible rank triplets, we select the one minimizing
$$\Delta(r_1, r_2, r_3) = |P_{\text{tucker}}(r_1, r_2, r_3) - P_{\text{target}}|.$$
This procedure reduces the original three-dimensional integer search to a two-dimensional sweep with a closed-form update for the third rank, enabling efficient budget-constrained rank allocation.

---

**Algorithm 2** Rank Allocation Algorithm

---

**Require:** MoE tensor size $(K, d_{\text{out}}, d_{\text{in}})$; target compression ratio $\rho^* \in (0, 1)$
**Ensure:** Tucker ranks $(r_1^*, r_2^*, r_3^*)$
1: $P_{\text{orig}} \leftarrow K d_{\text{out}} d_{\text{in}}$
2: $P_{\text{target}} \leftarrow (1 - \rho^*) P_{\text{orig}}$
3: best_err $\leftarrow +\infty$
4: $(r_1^*, r_2^*, r_3^*) \leftarrow (1, 1, 1)$
5: **for** $r_1 = 1, 2, \ldots, K$ **do**
6:     **for** $r_2 = 1, 2, \ldots, d_{\text{out}}$ **do**
7:         num $\leftarrow P_{\text{target}} - (K r_1 + d_{\text{out}} r_2)$
8:         den $\leftarrow r_1 r_2 + d_{\text{in}}$
9:         **if** num $\le 0$ **then**
10:             **continue**
11:         **end if**
12:         $r_3 \leftarrow \lfloor \text{num}/\text{den} \rfloor$
13:         **if** $r_3 < 1$ **or** $r_3 > d_{\text{in}}$ **then**
14:             **continue**
15:         **end if**
16:         $P_{\text{tucker}} \leftarrow r_1 r_2 r_3 + K r_1 + d_{\text{out}} r_2 + d_{\text{in}} r_3$
17:         err $\leftarrow |P_{\text{tucker}} - P_{\text{target}}|$
18:         **if** err $<$ best_err **then**
19:             best_err $\leftarrow$ err
20:             $(r_1^*, r_2^*, r_3^*) \leftarrow (r_1, r_2, r_3)$
21:         **end if**
22:     **end for**
23: **end for**
24: **return** $(r_1^*, r_2^*, r_3^*)$

---

For layerwise budget allocation, we follow MoE-SVD (Li et al., 2025) and assign larger parameter budgets to layers with higher expert-activation frequency, reflecting their greater functional contribution during routing. Let $a_\ell$ denote the expert-activation frequency score of layer $\ell$. We normalize the scores as
$$\widehat{a}_\ell = \frac{a_\ell}{\sum_{j=1}^{L} a_j}.$$
The target parameter budget for layer $\ell$ is then
$$P_\ell^* = \widehat{a}_\ell (1 - \rho^*) \sum_{j=1}^{L} P_{\text{orig}}^{(j)}.$$
This provides a simple and low-overhead mechanism for distributing the global compression budget while accounting for layerwise routing importance.

# D  STRUCTURED TUCKER PRUNING

Below we describe the post-training structured pruning in Section 4.3 applied after Tucker decomposition to remove redundant latent dimensions. Starting from a Tucker model with core tensor $\mathcal{G} \in \mathbb{R}^{r_1 \times r_2 \times r_3}$ and factor matrices $\{U_1, U_2, U_3\}$, we operate entirely in the latent space: no access to the original dense weights is required. The key idea is to exploit the energy distribution inside the core tensor. Each mode-2 slice $\mathcal{G}_{:,i,:}$ corresponds to one latent feature direction in the output mode, and each mode-3 slice $\mathcal{G}_{:,:,j}$ corresponds to one latent direction in the input mode. We measure the contribution of each slice via its $\ell_2$ norm (energy), rank slices by energy, and treat the lowest energy slices as redundant. Low-energy slices contribute little to the overall reconstruction and can therefore be safely removed, together with their associated columns in the factor matrices $U_2$ and $U_3$. Concretely, given a pruning ratio $\rho$, we retain only the top $(1 - \rho)$ fraction of slices along modes 2 and 3, and shrink the core and factor matrices accordingly. This yields a strictly smaller Tucker model with reduced ranks $(r_2', r_3')$, while preserving the multilinear structure and the original expert mode $U_1$. Because pruning is performed on top of an already compressed decomposition, it can be applied incrementally to trade parameters for accuracy with smooth, monotonic degradation, and does not require re-running Tucker. The full pruning pipeline is summarized in Algorithm 3.

---

**Algorithm 3** Structured Tucker Pruning for TD-MoE

---

**Require:** Core tensor $\mathcal{G} \in \mathbb{R}^{r_1 \times r_2 \times r_3}$, factor matrices $\{U_1, U_2, U_3\}$, pruning ratio $\rho \in [0, 1)$
**Ensure:** Pruned core $\mathcal{G}'$ and factors $\{U_1', U_2', U_3'\}$
 1: **if** $\rho \leq 0$ **then**
 2:     **return** $\mathcal{G}, \{U_1, U_2, U_3\}$
 3: **end if**
 4: $r_2' \leftarrow \max(1, \lfloor (1 - \rho)r_2 \rfloor), \quad r_3' \leftarrow \max(1, \lfloor (1 - \rho)r_3 \rfloor)$
 5: $e^{(2)} \leftarrow \|\mathcal{G}\|_{(1,3)} \in \mathbb{R}^{r_2}, \quad I_2 \leftarrow \text{TopK}(e^{(2)}, r_2')$
 6: $e^{(3)} \leftarrow \|\mathcal{G}\|_{(1,2)} \in \mathbb{R}^{r_3}, \quad I_3 \leftarrow \text{TopK}(e^{(3)}, r_3')$
 7: $\mathcal{G}' \leftarrow \mathcal{G}[:, I_2, I_3]$
 8: $U_1' \leftarrow U_1, \quad U_2' \leftarrow U_2[:, I_2], \quad U_3' \leftarrow U_3[:, I_3]$
 9: **return** $\mathcal{G}', \{U_1', U_2', U_3'\}$

---

# E  ADDITIONAL EXPERIMENTS

## E.1  RESULTS ON PHI-3.5-MOE

Table 6 presents results on Phi-3.5-MoE across 20%, 40%, and 60% compression. TD-MoE consistently ranks among the strongest methods. At 20–40% compression, TD-MoE closely matches the original model on language modeling metrics and achieves the highest reasoning accuracy among all baselines. Under aggressive 60% compression, a regime where most decompositions collapse, TD-MoE remains significantly more stable and preserves far better accuracy, highlighting its resilience under extreme compression.

Table 6: Performance on Phi-3.5-MoE.

| Ratio | Method | W2 | PTB | C4 | Reason Avg |
|---|---|---|---|---|---|
| origin | baseline | 3.5 | 8.4 | 8.2 | 0.62 |
| 20% | ASVD (Yuan et al., 2023) | 7.2 | 10.7 | 9.6 | 0.56 |
| | SVD-LLM (Wang et al., 2024) | 8.3 | 14.8 | 12.9 | 0.51 |
| | MoE-SVD (Li et al., 2025) | 4.6 | 10.1 | 9.9 | 0.60 |
| | TD-MoE | 4.7 | 9.2 | 9.1 | 0.61 |
| 40% | MoE-I2 (Yang et al., 2024) | 7.5 | 21.0 | 21.0 | 0.45 |
| | SVD-LLM (Wang et al., 2024) | 38.8 | 68.5 | 43.8 | 0.43 |
| | MoE-SVD (Li et al., 2025) | 5.5 | 11.7 | 11.9 | 0.56 |
| | TD-MoE | 6.4 | 10.9 | 10.8 | 0.58 |
| 60% | ASVD (Yuan et al., 2023) | 107.7 | 208 | 161 | 0.35 |
| | SVD-LLM (Wang et al., 2024) | 7168 | 7101 | 7119 | 0.31 |
| | MoE-SVD (Li et al., 2025) | 7.5 | 21.0 | 21.9 | 0.49 |
| | TD-MoE | 13.7 | 19.7 | 17.9 | 0.50 |

## E.2 Results on HumanEval and XNLI

We evaluate TD-MoE on code generation (HumanEval (Chen et al., 2021)) and multilingual NLI (XNLI (Conneau et al., 2018)). As shown in Table 7, TD-MoE improves HumanEval Pass@1 and XNLI accuracy while better preserving code-related and multilingual reasoning capabilities.

Table 7: Results on HumanEval (Pass@1) and XNLI.

| Task | Ratio | MoE-SVD | TD-MoE | Δ |
|------|-------|---------|--------|---|
| HumanEval (Pass@1) | Origin | – | 0.293 | – |
| HumanEval (Pass@1) | 0.2 | 0.079 | 0.240 | **+0.161** |
| HumanEval (Pass@1) | 0.4 | 0.037 | 0.140 | **+0.103** |
| XNLI (Acc.) | Origin | – | 0.450 | – |
| XNLI (Acc.) | 0.2 | 0.390 | 0.440 | **+0.050** |
| XNLI (Acc.) | 0.4 | 0.380 | 0.410 | **+0.030** |

