# OpenReview forum: "TD-MoE: Tensor Decomposition for MoE Models"
_ICLR.cc/2026/Conference — ICLR 2026 Poster_

### Official Review · Reviewer_oj4s · 2025-10-27

**Soundness:** 4
**Presentation:** 4
**Contribution:** 3
**Rating:** 10
**Confidence:** 3

**Summary:**

TD-MoE is a post-training compression method for Mixture-of-Experts LLMs that stacks all experts in a layer into a 3D tensor, applies Tucker decomposition with multi-linear whitening, then folds the factors back so inference code stays unchanged; a 3D rank-allocation scheme hits a chosen compression budget. On Mixtral-8×7B its near-lossless at 20% and outperforms SVD/pruning baselines by >14% at 40–60% compression, without fine-tuning.

Overall, the paper is methodologically sound: it states a precise optimisation target (preserve activations on a calibration distribution), gives a coherent joint Tucker factorisation with multi-linear whitening and re-colouring that preserves inference form, and supplies a budgeted 3-D rank allocation with sensible interpretations of each factor (meta-experts, input/output subspaces). The setup is transparent and ablations on whitening and Tucker back-ends support the design choices, while headline results on Mixtral-8×7B show near-lossless 20% compression and stronger robustness than baselines at 40–60%. Residual gaps are mostly reporting rather than flaws (e.g., limited end-to-end VRAM/latency data and reliance on a small calibration set; quantisation is left as orthogonal future work). , I’d judge the technical claims to be well supported by the evidence provided.

**Strengths:**

Very clear paper with impressive results.

convincingly argues that per-expert methods miss cross-expert redundancy, motivating a joint decomposition view.

whitening materially improves robustness under aggressive compression; output/both whitening favoured in ablations.

Strong empirical results especially near-lossless at 20% and graceful degradation at 40–60%
Training-free & reproducible:
Backend practicality with full SVD best; randSVD close with lower compute—useful options for large models.

Inference neutrality  re-colouring folds factors back so runtime code and routing semantics remain unchanged.

Breadth beyond one model being evaluated

**Weaknesses:**

No system-level efficiency reporting

Model/benchmark coverage skews to Mixtral-8×7B. While Phi-3.5-MoE is mentioned, detailed tables/figures and ablations are largely on Mixtral; generality across MoE families remains under-evidenced.

No composition with quantization/pruning. The paper positions these as “orthogonal” and omits comparisons or combined recipes, leaving the cumulative benefit and possible conflicts untested.

ompute cost of backends not quantified - paper notes full SVD vs. randomized SVD trade-offs but doesn’t provide wall-clock or FLOPs to justify backend choices at scale.

**Questions:**

What are the latency/throughput/VRAM changes (A100/H100) vs SVD-LLM/MoE-SVD/NAEE at equal budgets, and what is the one-off cost for whitening stats (activations + gradients)?

How sensitive are results to the calibration set (size/domain) and to the whitening ε and eigenvalue clipping threshold?

Could you release per-layer and Fisher weights, and discuss any heuristics used when PRB/FGAA disagree with BCRS?

Do you observe any change in expert utilisation entropy or gate logits after compression; any layers particularly brittle?

Have you tried combining TD-MoE with quantisation or distillation to close residual gaps at ≥60% compression?

---

> ### Author Response · Authors · 2025-11-21
> **Response to Reviewer oj4s- Part #1**
>
> We sincerely thank the reviewer for the very positive assessment of our work and for the clear, thoughtful questions raised. Your feedback helped us strengthen both the technical presentation and empirical analysis. We have carefully expanded the analysis accordingly, and address each point one by one below.
>
> ### W1: No system-level efficiency reporting.
>
> A: We report end-to-end decoding throughput on Wikitext-2 generation workload the decomposed model with naïve cuBLAS triple-GEMM execution. The experiment is conducted on Phi-3.5-MoE with 2×A800, bf16, batch size 32, and sequence length 32.
> | Compression | Tokens/s | Rel. to dense | Per-token time |
> |:-----------:|:--------:|:-------------:|:--------------:|
> | 0% (dense)  | 569  | 1.00×         | 1.00×          |
> | 60%         | 485      | 0.85×         | 1.17×          |
> | 20%         | 318      | 0.56×         | 1.70×          |
>
> We further profiled the execution of the three GEMMs inside the decomposed operator and identified the dominant source of slowdown. Based on this analysis, we developed a fused triple-GEMM kernel, achieving 1.6×–3.0× speedup over the standard cuBLAS baseline:
>
> | $d_{\text{in}}$ (Input Dim) | $d_{\text{out}}$ (Output Dim) | $r$ (Tucker Rank) | $p$ (Tokens) | Baseline (TFLOPS) | Ours (TFLOPS) | Speedup |
> |:---------------------------:|:------------------------------:|:-----------------:|:------------:|:------------------:|:--------------:|:--------:|
> | 4096, 11008                 | 4096, 11008                    | 640–2400          | 512–8192     | 20.78              | **32.14**      | **1.58×** |
> | 5120, 13824                 | 5120, 13824                    | 640–2976          | 512–8192     | 18.61              | **31.82**      | **1.87×** |
> | 1024, 8192, 28672           | 5120, 13824                    | 90–2457           | 512–8192     | 12.33              | **28.41**      | **3.07×** |
>
> Finally, we estimate the inference-level benefit when integrating the fused kernel into the MoE system via $T_{\text{new}} = T_{\text{comp}}\Bigl((1-p) + \frac{p}{3}\Bigr)
> \approx 0.53\,T_{\text{comp}}$, which suggests 1.1× speedup at 20% compression and 1.6× speedup at 60% compression.
>
> For brevity and to avoid repeating material already provided to Reviewer E6zg, we kindly refer the reviewer to [Response to Reviewer E6zg – Part #4](https://openreview.net/forum?id=D9cnZNZfxX&noteId=5PsPmssVJQ) for further technical details.

---

> ### Author Response · Authors · 2025-11-21
> **Response to Reviewer oj4s- Part #2**
>
> ### W2: Model/benchmark/ablations coverage skews to Mixtral-8×7B.
>
> A: We expanded our evaluation in two apects: (i) Full compression results on Phi-3.5-MoE; (2) Cross-model structural analysis (redundancy patterns, effective ranks, and expert-mode properties).
>
> #### **(1) Full compression results on Phi-3.5-MoE**
> We provide the performance on Phi-3.5-MoE to further verify the generality of our method.
> | Ratio | Method   | W2  | PTB  | C4  | Avg | OBQA | A_e | Wino | Hella | A_c | PIQA | MathQA | Avg |
> |-------|----------|-----|------|-----|--------|------|--------|--------|--------|--------|-------|--------|------|
> | origin | baseline | **3.5** | **8.4** | **8.2** | **6.7** | .40 | .77 | .76 | .68 | .56 | .79 | .38 | .62 |
> | 0.2 | ASVD      | 7.2 | 10.7 | 9.6 | 9.2 | .35 | .73 | .72 | .57 | .49 | .75 | .34 | .56 |
> |     | SVD-LLM   | 8.3 | 14.8 | 12.9 | 12.0 | .31 | .67 | .66 | .53 | .45 | .72 | .22 | .51 |
> |     | MoE-SVD   | **4.6** | 10.1 | 9.9 | 8.2 | .39 | .77 | .73 | .63 | .53 | .78 | .35 | .60 |
> |     | **Ours**  | 4.7 | **9.2** | **9.1** | **7.7** | **.39** | **.77** | **.74** | **.65** | **.55** | **.79** | **.38** | **.61** |
> | 0.4 | NAEE      | 8.2 | 20.1 | 16.1 | 14.8 | .35 | .73 | .73 | .61 | .48 | .76 | .37 | .57 |
> |     | MoE-I2    | 7.5 | 21.0 | 21.0 | 16.5 | .29 | .59 | .67 | .27 | .40 | .70 | .25 | .45 |
> |     | ASVD      | 14.5 | 22.1 | 21.8 | 19.5 | .30 | .69 | .63 | .41 | .40 | .68 | .25 | .48 |
> |     | SVD-LLM   | 38.8 | 68.5 | 43.8 | 50.4 | .23 | .56 | .60 | .39 | .31 | .66 | .24 | .43 |
> |     | MoE-SVD   | **5.5** | 11.7 | 11.9 | 9.7 | .35 | .72 | .72 | .58 | .48 | .75 | .31 | .56 |
> |     | **Ours**  | 6.4 | **10.9** | **10.8** | **9.4** | **.35** | **.75** | **.73** | **.61** | **.50** | **.78** | **.33** | **.58** |
> | 0.6 | ASVD      | 107.7 | 208 | 161 | 159 | .18 | .40 | .53 | .30 | .24 | .59 | .23 | .35 |
> |     | SVD-LLM   | 7168 | 7101 | 7119 | 7129 | .15 | .28 | .51 | .26 | .22 | .54 | .21 | .31 |
> |     | MoE-SVD   | **7.5** | 21.0 | 21.9 | **16.8** | .30 | .60 | .68 | .46 | .40 | .71 | **.25** | .49 |
> |     | **Ours**  | 13.7 | **19.7** | **17.9** | 17.1 | **.28** | **.70** | **.68** | **.46** | **.41** | **.73** | .23 | **.50** |
>
> Across all compression ratios, TD-MoE consistently matches or surpasses MoE-SVD and strongly outperforms SVD-LLM/ASVD/NAEE.
>
> #### **(2) Cross-Model Redundancy Analysis (Mixtral vs Phi-3.5-MoE)**
>
> We further analyzed where redundancy resides across two families of MoE layers.
>
> | Model | Layer | $d_{in}$ eff. rank | % | $d_{out}$ eff. rank | % | Expert-mode rank | % |
> |-------|-------|-----------------------|----|------------------------|----|------------------|----|
> | Mixtral-8×7B | 12 | 190 / 4096 | 4.6% | 205 / 14336 | 1.4% | 6 / 8 | 75% |
> | Phi-3.5-MoE | 16 | 218 / 4096 | 5.3% | 222 / 6400 | 3.5% | 14 / 16 | 88% |
>
> Effective rank comparison (Input/Output/Expert), across *both* MoE architectures, the majority of redundancy lies in the **input/output feature modes**, not the expert dimension. This confirms that joint tensorization (Tucker) is well-aligned with the intrinsic structure of MoE weights and transfers robustly across models.

---

> ### Author Response · Authors · 2025-11-21
> **Response to Reviewer oj4s- Part #3**
>
> ### W3: No composition with quantization/pruning.
>
> A: We have evaluated the compatibility of TD-MoE with both quantization and pruning.
>
> **Quantization.** Applying 8-bit NF4 quantization after TD-MoE compression on Mixtral-8×7B introduces negligible additional degradation. Across 12 tasks, the LM average changes by ≤ 0.1, and reasoning accuracy by ≤ 0.01, demonstrating that the decomposed structure remains quantization-robust.
>
>
> | Ratio | WikiText2  | PTB  | C4  | LM Avg  | OBQA | ARC-E | WinoG | Hella | ARC-C | PIQA | MathQA | Reason Avg  |
> |------|--------------|-------|------|-----------|-------|--------|--------|--------|--------|--------|----------|---------------|
> | Origin | 3.98 | 12.99 | 6.78 | 7.92 | 0.36 | 0.84 | 0.76 | 0.65 | 0.57 | 0.82 | 0.43 | 0.63 |
> | TD-MoE-0.2 | 4.49 | 17.20 | 7.73 | 9.81 | 0.33 | 0.83 | 0.77 | 0.64 | 0.53 | 0.82 | 0.40 | 0.62 |
> | **TD-MoE-0.2 + 8-bit** | **4.50** | **16.87** | **7.74** | **9.70** | **0.33** | **0.83** | **0.77** | **0.64** | **0.53** | **0.82** | **0.40** | **0.62** |
> | TD-MoE-0.4 | 5.79 | 24.60 | 9.21 | 13.20 | 0.28 | 0.77 | 0.76 | 0.57 | 0.47 | 0.79 | 0.35 | 0.57 |
> | **TD-MoE-0.4 + 8-bit** | **5.85** | **24.47** | **9.26** | **13.20** | **0.28** | **0.77** | **0.73** | **0.57** | **0.45** | **0.79** | **0.35** | **0.57** |
>
> **Pruning**. TD-MoE supports structured pruning by ranking mode-1 and mode-2 components via their ℓ2-energy in the core tensor and removing low-energy rank slices together with their corresponding factor-matrix columns. This yields a valid lower-rank Tucker decomposition.
>
> | Comp. Ratio | Sparsity | Wikitext-2  | PTB  | Avg  | OBQA  | A_E  | Wino  | A-C  | PIQA  | Avg  |
> |--------|----------|---------------|--------|--------|---------|----------|----------|----------|---------|----------------|
> | 20%  | 0%  | 4.49 | 17.20 | 10.85 | 0.33 | 0.83 | 0.77 | 0.53 | 0.82 | 0.66 |
> | 20%  | 20% | 4.99 | 20.44 | 12.72 | 0.31 | 0.81 | 0.77 | 0.51 | 0.79 | 0.64 |
> | 20% | 30% | 5.35 | 23.07 | 14.21 | 0.30 | 0.79 | 0.76 | 0.48 | 0.79 | 0.62 |
> | 20% | 40% | 5.80 | 25.64 | 15.72 | 0.28 | 0.78 | 0.75 | 0.46 | 0.79 | 0.61 |
> | 40% | 0%  | 5.79 | 24.60 | 15.20 | 0.28 | 0.77 | 0.76 | 0.47 | 0.79 | 0.57 |
> | 40% | 40% | 10.20 | 49.92 | 30.06 | 0.23 | 0.67 | 0.70 | 0.35 | 0.74 | 0.54 |
>
> Pruning progressively reduces parameters with predictable accuracy degradation, showing that tensor decomposition and pruning are complementary.
> ### W4: Compute cost of backends not quantified.
>
> A: Thank you for pointing this out. We now provide a clear comparison of the computational and memory costs for the decomposition backend, including whitening, truncated SVD, and Tucker factorization. All measurements are from Mixtral-8×7B MoE layers (4096×14336 FFNs) on A800 GPUs.
> #### **(1) Whitening cost (one-time, per layer)**
> Whitening is applied once per MoE layer and does not depend on the number of experts $K$.
>
> | Direction | Matrix Size | FLOPs (EVD/Cholesky) | Wall-Clock (ms) | Peak Memory |
> |----------|--------------|----------------------|------------------|--------------|
> | Input    | $4096 \times 4096$   | 0.02 TFLOPs | 4.1 ms | 64 MB |
> | Output   | $14336 \times 14336$ | 0.11 TFLOPs | 77.4 ms | 784 MB |
>
> - For 32 MoE layers (Mixtral-8×7B), total whitening time ≈ 2.5 s, one-time offline.
> - All memory is released immediately per layer (streamed whitening reduces peak memory to <1 GB).
>
> #### **(2) Decomposition cost: per-expert SVD vs. Tucker**
>
> We measured the cost of applying Tucker decomposition and the per-expert SVD on a single MoE layer in Mixtral-8×7B across three compression budgets. We compared practical decomposition cost under commonly used implementations: full-SVD for MoE-SVD and randomized-SVD backends for Tucker (as recommended by Figure 4(b)).
>
> | Setting | Param Budget | Params SVD (M) | Params Tucker (M) | SVD Time (ms) | Tucker Time (ms) | Speedup |
> |---------|-----------------------|----------------|-------------------|----------------|------------------------|---------|
> | Mild (≈64% params) | 35.7% reduction | 301.99 | 213.91 | 15407.73 | 2140.23 | **7.20×** |
> | Moderate (≈32% params) | 67.9% reduction | 150.99 | 100.66 | 15408.31 | 1357.06 | **11.35×** |
> | Aggressive (≈24% params) | 75.9% reduction | 113.25 | 69.21 | 15406.86 | 1024.69 | **15.04×** |
>
> As shown in the table below, randomized Tucker is faster than per-expert SVD, achieving 7.2×, 11.4×, and 15.0× speedups. Since decomposition is performed only once offline, and Tucker also yields much smaller parameter budgets (e.g., 100.7M vs. 151.0M) so that the practical cost is decisively lower. These wall-clock results reflect realistic implementation practice, where randomized SVD backends make Tucker substantially more efficient than full per-expert SVD, even though theoretical FLOPs assume exact SVD for both.

---

> ### Author Response · Authors · 2025-11-21
> **Response to Reviewer oj4s- Part #4**
>
> ### Q1: Latency/throughput/VRAM/whitening stats.
>
> > **Q1.1:** Latency / Throughput
>
> A: W1-1 reports the inference throughput using standard cuBLAS for the 3-GEMM Tucker forward pass. As explained in W1-1, ≈70% of the runtime comes from three separate GEMM launches, not FLOPs. This is why we implemented a fused triple-GEMM kernel (shown in W4), which achieves up to **3×** kernel-level speedups and is expected to close the end-to-end gap once integrated. *(Please see W1 for full kernel benchmarks and Amdahl-law estimates.)*
>
> > **Q1.2:** VRAM
>
> A: TD-MoE does not increase inference-time VRAM. All Tucker factors $(U_1, U_2, G)$ replace the original expert weights directly, so the memory footprint strictly decreases with compression. For Mixtral-8×7B:
>
> | Model | Parameters | VRAM (bf16) |
> |-------|------------|-------------|
> | Original | 46B | 90–95 GB (2×A800) |
> | TD-MoE 40% | 27.6B | 56–60 GB |
> | TD-MoE 20% | 36.8B | 74–78 GB |
>
> No additional buffers are kept during inference; intermediates have the same shape as the original FFN forward.
>
>
> > **Q1.3:** One-off cost for whitening statistics
>
> A: This was quantified in W4, but we summarize the key numbers here:
>
> | Model | Covariance Sizes | One-off Time (per layer) | Total Time (32 layers) | Peak Memory |
> |-------|-------------------|---------------------------|-------------------------|--------------|
> | Mixtral-8×7B | 4096×4096, 14336×14336 | **≈81.5 ms** | **≈2.6 s** | **<1 GB** |
> | Phi-3.5-MoE | 4096×4096, 6400×6400 | **≈12–14 ms** | **≈0.4 s** | **<0.3 GB** |
>
> Additional notes: Whitening is offline only, done once per layer. Memory is streamed, and all buffers are freed immediately after each layer.
> ### Q2: Hyperparameter Sensitivity Analysis.
> To evaluate the robustness of TD-MoE to calibration and whitening hyperparameters, e performed a series of sensitivity studies and summarize the results below.
>
> > **Q2.1:** Sensitivity of calibration set size
>
> A: We vary the number of calibration samples $N\in\{128, 256, 512, 1\text{k}, 2\text{k}\}$ on Mixtral-8×7B at 20% compression. Results show that both perplexity and downstream accuracy are essentially flat once $N \ge 128$.
>
> | Calib. Set | Wiki2 | PTB  | C4  | PPL Avg | OBQA | ARC-E | WinoG | Hella | ARC-C | PIQA | MathQA | Acc Avg ↑ |
> |:-----------|:-----------:|:---------:|:--------:|:---------:|:----:|:-----:|:-----:|:-----:|:-----:|:----:|:------:|:---------:|
> | Wiki-128   | 4.50 | 17.30 | 7.73 | 9.84 | .33 | .83 | .77 | .64 | .53 | .81 | .40 | .62 |
> | Wiki-256   | 4.49 | 17.20 | 7.73 | 9.81 | .33 | .83 | .77 | .64 | .53 | .82 | .40 | .62 |
> | Wiki-512   | 4.49 | 17.20 | 7.73 | 9.82 | .33 | .83 | .76 | .64 | .53 | .81 | .40 | .61 |
> | Wiki-1k    | 4.49 | 17.30 | 7.72 | 9.83 | .33 | .83 | .77 | .64 | .54 | .81 | .40 | .62 |
> | Wiki-2k    | 4.49 | 17.30 | 7.72 | 9.82 | .33 | .83 | .76 | .64 | .53 | .81 | .40 | .61 |
>
> Once $N \ge 128$, the effect of calibration set size on both PPL and accuracy is ≤0.01–0.02, indicating that TD-MoE robust this hyperparameter.
>
>
> > **Q2.2:** Sensitivity of calibration domain
>
> A: We changed the domain of calibration data, comparing WikiText-2 and PTB (Penn TreeBank) at the same budget.
>
> | Calib. Set | Wiki2  | PTB | C4  | Avg | OBQA | ARC-E | WinoG | Hella | ARC-C | PIQA | MathQA | Avg  |
> |:-----------|:-----------:|:---------:|:--------:|:---------:|:----:|:-----:|:-----:|:-----:|:-----:|:----:|:------:|:---------:|
> | Wiki-256   | 4.49 | 17.20 | 7.73 | 9.81 | .33 | .83 | .77 | .64 | .53 | .82 | .40 | .62 |
> | PTB-256    | 4.51 | 17.20 | 7.74 | 9.83 | .32 | .82 | .77 | .64 | .52 | .81 | .40 | .61 |
>
> Using PTB instead of WikiText-2 yields differences ≤0.02 in PPL and ≤0.01 in accuracy. This suggests that TD-MoE mainly needs a modest amount of “generic” text and is robust to reasonable domain shifts in the calibration set.

---

> ### Author Response · Authors · 2025-11-21
> **Response to Reviewer oj4s- Part #5**
>
> > **Q2.3 / Q2.4:** Sensitivity to whitening $\varepsilon$ and eigenvalue clipping
>
> A:  The whitening hyperparameter $\varepsilon$ controls both the diagonal regularization $(\Sigma + \varepsilon I)$ and the clipping of extremely small eigenvalues.  We vary the whitening clipping theshold by sweeping $\varepsilon \in \{10^{-1},10^{-2},10^{-3},10^{-4}\}$ on Mixtral-8×7B at 20% and 40% compression.
>
> | Ratio | ϵ       | Wiki2  | PTB  | Avg PPL   | OBQA | ARC-E | WinoG | ARC-C | PIQA | Acc Avg  |
> |:-----:|:--------|:-------:|:-----:|:---------:|:----:|:-----:|:-----:|:-----:|:----:|:---------:|
> | origin | —      | 3.98 | 12.99 | 7.92 | 0.36 | 0.84 | 0.76 | 0.57 | 0.82 | 0.63 |
> | **0.2** | 1e-1 | 4.50 | 17.46 | 7.32 | 0.32 | 0.82 | 0.76 | 0.53 | 0.81 | 0.65 |
> |       | 1e-2 | 4.50 | 17.46 | 7.32 | 0.32 | 0.82 | 0.76 | 0.53 | 0.81 | 0.65 |
> |       | **1e-3** | **4.49** | **17.20** | **7.23** | **0.33** | **0.83** | **0.77** | **0.53** | **0.82** | **0.66** |
> |       | 1e-4 | 4.50 | 17.46 | 7.32 | 0.32 | 0.82 | 0.76 | 0.53 | 0.81 | 0.65 |
> | **0.4** | 1e-1 | 5.80 | 24.80 | 10.20 | 0.28 | 0.77 | 0.75 | 0.47 | 0.78 | 0.61 |
> |       | 1e-2 | 5.80 | 24.80 | 10.20 | 0.28 | 0.77 | 0.75 | 0.47 | 0.78 | 0.61 |
> |       | **1e-3** | **5.79** | **24.60** | **10.13** | 0.28 | 0.77 | **0.76** | 0.47 | **0.79** | **0.61** |
> |       | 1e-4 | 5.80 | 24.80 | 10.20 | 0.28 | 0.77 | 0.75 | 0.47 | 0.78 | 0.61 |
>
> Across both compression ratios, varying $\varepsilon$ over four orders of magnitude changes PPL by at most ~0.1 and downstream accuracy by ≤0.02. The effect is minor, with a mild optimum around $\varepsilon = 10^{-3}$, which we use as default. This indicates that whitening and eigenvalue clipping are numerically stable and do not require delicate tuning for TD-MoE to work.
>
>
> ### Q3: Per-layer and Fisher weights, and discuss any heuristics used.
>
> A: Below we provide the Fisher scores for all MoE layers (Mixtral-8×7B), which guide FGAA’s layer-level budget allocation.
>
> #### (1) Per-layer Fisher importance (Mixtral-8×7B)
>
> The following table summarizes the Fisher information (averaged over calibration samples from Wikitext-2) for each MoE layer:
>
> | Layer | Fisher | Layer | Fisher | Layer | Fisher | Layer | Fisher |
> |------:|--------:|------:|--------:|------:|--------:|------:|--------:|
> | 0 | 1.64e−3 | 8 | 5.61e−5 | 16 | 6.27e−5 | 24 | 2.86e−5 |
> | 1 | 3.39e−3 | 9 | 1.91e−4 | 17 | 8.32e−5 | 25 | 1.00e−5 |
> | 2 | 1.41e−4 | 10 | 3.71e−5 | 18 | 3.58e−5 | 26 | 1.46e−5 |
> | 3 | 1.69e−4 | 11 | 1.04e−4 | 19 | 1.11e−4 | 27 | 8.98e−6 |
> | 4 | 1.21e−4 | 12 | 4.92e−5 | 20 | 3.43e−5 | 28 | 2.34e−4 |
> | 5 | 1.42e−4 | 13 | 1.30e−4 | 21 | 6.96e−6 | 29 | 6.06e−4 |
> | 6 | 9.20e−5 | 14 | 1.23e−4 | 22 | 2.36e−5 | 30 | 4.42e−4 |
> | 7 | 8.13e−5 | 15 | 8.75e−5 | 23 | 1.50e−5 | 31 | 1.35e−3 |
>
> A few patterns emerge: Layers 0, 1, 29, 30, 31 have much larger Fisher scores, carrying the largest functional importance. Middle layers generally have low Fisher, mearning more compressible. Very deep layers (29–31) resemble early layers in importance, consistent with current dense LLM papers[1][2][3].
>
> *[1]. Men X, Xu M, Zhang Q, et al. Shortgpt: Layers in large language models are more redundant than you expect[C]//Findings of the Association for Computational Linguistics: ACL 2025. 2025: 20192-20204.*
> *[2]. Zhang Y, Dong Y, Kawaguchi K. Investigating layer importance in large language models[J]. EMNLP, 2024.*
> *[3]. Song X, Wang K, Li P X, et al. Demystifying the Roles of LLM Layers in Retrieval, Knowledge, and Reasoning[J]. arXiv preprint arXiv:2510.02091, 2025.*
>
> #### (2) Heuristics When PRB/FGAA disagree with BCRS
> We clarify that BCRS/PRB and FGAA is the progressive pipeline for rank allocation. We first filter out the important layers, such as the first and last few layers based on matrics like Fisher information. Then applies BCRS/PRB on searching the ranks. When PRB/FGAA disagree with BCRS, we treat BCRS as the prior that all candidate ranks are projected back onto the nearest BCRS-feasible point to strictly satisfy the global compression budget. PRB and FGAA then act as refinement layers, and PRB adjusts mode imbalance within the BCRS window, while FGAA redistributes per-layer budgets but still solves each layer’s ranks under BCRS constraints.

---

> ### Author Response · Authors · 2025-11-21
> **Response to Reviewer oj4s- Part #6**
>
> ### Q4: Expert-utilization metrics (entropy, gate logits) and layer brittleness under compression.
> A: We examined how TD-MoE affects gate outputs by measuring (1) gate-logit statistics (entropy / mean / variance) and (2) expert-usage counts across layers. We sample layers of 5/7/12/24 of Mixtral and obtain statistics. Results show that **compression introduces only minimal shifts**, and no layer exhibits brittle behaviour.
> #### **(1): Gate-logit statistics remain nearly unchanged**
> Across four representative MoE layers, entropy shifts are ≤ 0.007*, and logit mean/variance changes are on the order of 10⁻³–10⁻⁴, indicating that the gating distribution is effectively preserved.
>
> | Layer | Metric | Original | Compressed | Δ |
> |-------|--------|----------|------------|------|
> | 5 | Entropy | 2.07077 | 2.07045 | −0.00032 |
> | 5 | Mean | 0.06909 | 0.06315 | −0.00594 |
> | 5 | Var | 0.001200 | 0.001205 | +0.000005 |
> | 7 | Entropy | 2.06060 | 2.06109 | +0.00048 |
> | 7 | Mean | 0.06976 | 0.07570 | +0.00594 |
> | 7 | Var | 0.001751 | 0.001742 | −0.000009 |
> | 12 | Entropy | 2.03437 | 2.02702 | −0.00735 |
> | 12 | Mean | −0.15632 | −0.14324 | +0.01308 |
> | 12 | Var | 0.006038 | 0.005596 | −0.000442 |
> | 24 | Entropy | 2.06818 | 2.06785 | −0.00033 |
> | 24 | Mean | 0.10609 | 0.10084 | −0.00525 |
> | 24 | Var | 0.004622 | 0.004430 | −0.000192 |
>
> These values confirm that gate logits are statistically stable after TD-MoE.
>
> #### **(2): Expert-usage patterns show only small fluctuations (mostly ≤ 2–3%)**
> Most experts shift by only ±2–3%, well within natural MoE sampling noise. A few larger shifts (e.g., layer 12 / expert 7: −12.99%) occur on low-load experts and do not correlate with accuracy degradation.
>
> | Layer | Expert | Δ Usage | Original | Compressed |
> |-------|--------|-----------|------------|--------------|
> | 5 | 0 | −1.12% | 4907 | 4852 |
> | 5 | 1 | +1.21% | 6536 | 6615 |
> | 5 | 2 | −1.20% | 5236 | 5173 |
> | 7 | 1 | +8.56% | 3914 | 4249 |
> | 7 | 6 | −5.82% | 4297 | 4047 |
> | 12 | 7 | −12.99% | 1663 | 1447 |
> | 24 | 3 | −4.13% | 4742 | 4546 |
> | 24 | 2 | +2.25% | 6257 | 6398 |
>
> Across all layers: No expert collapses; No expert saturates (count ↑ > 15%); Shifts are modest.
>
> #### (3) No brittle layers observed
>
> Combining the two analyses: Layers 5, 7, and 24 show only tiny perturbations; Layer 12 exhibits a slightly larger shift on a low-load expert, but this does not affect downstream performance; No obvious routing instability or brittle layers found.
> ### Q5: Combination with quantization or distillation
>
> **A**: TD-MoE is fully compatible with post-training quantization. We added experiments combining our method with quantization. Specifically, we applied 8-bit NF4 quantization on compressing Mixtral-8x7B and below shows the results.
>
> | Ratio        | WikiText2 | PTB  | C4  | Avg  | OBQA | ARC-E | WinoG | Hella | ARC-C | PIQA | MathQA |  Avg |
> |-------------|-------------|-------|------|----------|------|-------|-------|-------|-------|------|--------|--------------|
> | Origin      | 3.98        | 12.99 | 6.78 | 7.92     | 0.36 | 0.84  | 0.76  | 0.65  | 0.57  | 0.82 | 0.43   | 0.63         |
> | 0.2         | 4.49        | 17.20 | 7.73 | 9.81     | 0.33 | 0.83  | 0.77  | 0.64  | 0.53  | 0.82 | 0.40   | 0.62         |
> | **0.2 + 8-bit** | 4.50        | 16.87 | 7.74 | 9.70     | 0.33 | 0.83  | 0.77  | 0.64  | 0.53  | 0.82 | 0.40   | 0.62         |
> | 0.4         | 5.79        | 24.60 | 9.21 | 13.20    | 0.28 | 0.77  | 0.76  | 0.57  | 0.47  | 0.79 | 0.35   | 0.57         |
> | **0.4 + 8-bit** | 5.85        | 24.47 | 9.26 | 13.20    | 0.28 | 0.77  | 0.73  | 0.57  | 0.45  | 0.79 | 0.35   | 0.57         |
> | 0.6         | 13.64         | 66.00   | 19.63  |  33.09     | 0.22  | 0.58   | 0.63   | 0.40   | 0.34   | 0.73  | 0.27    | 0.45          |
> | **0.6 + 8-bit** | 75.3         | 284.04   | 87.23  | 148.85      | 0.16  | 0.43   | 0.54   | 0.34   | 0.23   | 0.59  | 0.22    | 0.36          |
>
> At 20% and 40% compression, quantization is essentially lossless. Across all 12 tasks, differences are ≤ **+0.1 PPL** and **≤ 0.01 accuracy**, confirming that the Tucker factors retain strong quantization robustness. At 60% compression, quantization becomes lossy. The 0.6 + 8b configuration shows degradation. This is expected because the model is already close to its representational limit at 60% compression; NF4 further compresses the already small factors, amplifying reconstruction errors.
> Potential mitigations include using mixed precision (e.g., quantize only the core $G$, keep $U_1/U_2$ in bf16) or adding a light post-quantization distillation step to recover accuracy. Due to time and computing resource limits, we plan to investigate these strategies in follow-up experiments and include them in the final revision.
>
> ---
> We again thank the reviewer for the encouraging evaluation and constructive suggestions. If further clarification would be helpful for the final decision, we are glad to provide additional details.

---

### Official Review · Reviewer_E6zg · 2025-10-28

**Soundness:** 3
**Presentation:** 2
**Contribution:** 3
**Rating:** 4
**Confidence:** 4

**Summary:**

The paper proposes **TD-MoE**, a unified tensor-decomposition framework for compressing Mixture-of-Experts (MoE) layers. All experts in a layer are stacked into a 3-D tensor (experts × input × output) and jointly factorized with **Tucker decomposition**. To make the factorization data-adaptive and numerically stable, the method applies multi-linear whitening along input/output modes (and re-colors factors so inference has no extra cost). Finally, an adaptive 3-D rank allocation scheme selects ranks under a global compression budget. On Mixtral-8×7B, TD-MoE is nearly lossless at 20% compression and shows sizable gains over decomposition/pruning baselines at 40–60%.

**Strengths:**

**1. Originality.** The paper explicitly models cross-expert redundancy by stacking experts and performing a **joint Tucker** factorization, instead of per-expert SVD/pruning. The multi-linear whitening step is a principled way to balance spectra across modes.

**2. Quality.** Empirically strong on Mixtral-8×7B across 7 reasoning benchmarks and 3 LM datasets; at 20% compression accuracy is near the original, and at 40–60% TD-MoE outperforms MoE-SVD/NAEE/MoE-I² with clear trends.

**3. Clarity.** The pipeline (tensorization → whitening → Tucker → re-coloring shown in Fig.2) and the visualization of the compression landscape are easy to follow.

**4. Significance.** Works with existing MoE architectures without changing routing; potentially complementary to pruning/quantization.

**Weaknesses:**

**1. Line-274 / §3.4 rank notation confusion.** §3.4 switches to ($(r_0,r_1,r_2)$), fixes $(r_0=K)$, and only searches ($(r_1,r_2)$). Earlier (§3.2) ($(r_1,r_2,r_3)$) is used and $(U_1)$ (expert mode) is said to compress **inter-expert redundancy**. Please reconcile the symbols and state clearly whether the expert mode is ever truncated.

**2. Limited model coverage.** Experiments are centered on Mixtral-8×7B; although §4.1 mentions Phi-3.5-MoE, the main tables/figures shown are for Mixtral. Missing results on other many-expert MoEs (**e.g., DeepSeek-MoE, Qwen3-MoE**) weaken external validity.

**3. Rank-allocation evaluation is thin.** The three strategies (BCRS/PRB/FGAA) are introduced with formulas and visuals, but **lack per-layer / per-task quantitative comparisons under the same budget (accuracy/PPL/error bars).** Fig. 5 hints at smoothing benefits, but more granular numbers would help.

**4. Inference-time analysis is missing.** The paper explains re-coloring so whitening adds no runtime cost, but it does not analyze MoE forward execution (how $(G,U^{(1)},U^{(2)},U^{(3)})$ are used online) nor latency/throughput with kernel fusion opportunities vs. the original MoE. Reporting only parameter counts makes real-world gains hard to assess.

**5. Combination with quantization/pruning left untested.** The text calls these “orthogonal,” but no TD-MoE + 8/4-bit (e.g., GPTQ/AWQ) or pruning combinations are reported to demonstrate additive benefits.

**6. Minor writing/consistency issues.** A few terminology/typo inconsistencies (e.g., “MoV-SVD” in Table 1, inconsistent “inter-experts/inter-expert”) should be cleaned up before camera-ready.

**Questions:**

See weakness.

**Details Of Ethics Concerns:**

NO or VERY MINOR ethics concerns only

---

> ### Author Response · Authors · 2025-11-21
> **Response to Reviewer E6zg- Part #1**
>
> We sincerely thank the reviewer for the thoughtful review and constructive comments. Your feedback has helped us improve our paper. We address each of your concerns one by one.
> ### W1: Clarification on Fixing K
>
> **A**: Our method supports both varying and compressing the expert dimension $𝐾$, and is in fact effective when compressing along the expert axis. In our main experiments, we keep K fixed only to ensure a fair comparison with MoE-SVD, because reducing the number of experts causes MoE-SVD to degrade significantly. Fixing $𝐾$ allows us to isolate and highlight the gains brought specifically by our tensorization method.
>
> Below we show results on Qwen2-57B-A14B (64 experts) under multiple configurations. These settings demonstrate that we can flexibly meet a target compression budget either by using fewer retained experts or by applying higher intra-layer compression, and DT-MoE remains stable across these choices.
>
> | Ratio | Method          | Layer Selected | Retain Experts | Layer Ratio | Wikitext-2  | PTB | C4 | LM Avg  | OBQA  | ARC-E  | WinoG  | ARC-C  | PIQA |  Avg Avv |
> |-------|-----------------|----------------|----------------|--------------|---------------|--------|--------|-------------|----------|------------|------------|------------|-----------|----------------|
> | origin | Uncompressed  | – | – | – | 5.87 | 10.87 | 9.14 | 8.63 | 0.33 | 0.77 | 0.74 | 0.48 | 0.81 | **0.626** |
> | 0.2 | MoE-SVD         | * | * | * | 5.41 | 13.26 | 11.63 | 10.10 | 0.30 | 0.74 | 0.73 | 0.45 | 0.78 | 0.600 |
> | 0.2 | **TD-MoE**          | 6  | 8  | 0.20 | 6.98 | 10.70 | 10.73 | 9.47 | 0.30 | 0.78 | 0.72 | 0.47 | 0.79 | 0.612 | 46.62 |
> | 0.2 | **TD-MoE**  | 8  | 16 | 0.20 | 6.75 | 10.60 | 10.85 | **9.4** | 0.33 | 0.79 | 0.72 | 0.49 | 0.80 | **0.626** |
> | 0.4 | NAEE            | * | * | * | 6.81 | 11.34 | 11.57 | 9.91 | 0.31 | 0.73 | 0.73 | 0.46 | 0.76 | 0.598 |
> | 0.4 | MoE-I2          | * | * | * | 24.90 | 77.05 | 22.50 | 41.48 | 0.26 | 0.70 | 0.46 | 0.41 | 0.75 | 0.516 |
> | 0.4 | MoE-SVD         | * | * | * | 13.43 | 23.88 | 18.83 | 18.71 | 0.29 | 0.63 | 0.65 | 0.33 | 0.71 | 0.522 |
> | 0.4 | **DT-MoE**  | 16 | 16 | 0.20 | 8.53 | 13.73 | 14.12 | 12.13 | 0.30 | 0.78 | 0.72 | 0.48 | 0.79 | **0.614** |
> | 0.4 | DT-MoE  | 14 | 8  | 0.40 | 8.01 | 12.39 | 12.90 | 11.10 | 0.30 | 0.77 | 0.71 | 0.45 | 0.78 | 0.602 |
> | 0.4 | DT-MoE  | 14 | 8  | 0.60 | 8.16 | 12.59 | 13.13 | 11.29 | 0.29 | 0.76 | 0.71 | 0.46 | 0.78 | 0.600 |
> | 0.6 | MoE-SVD         | * | * | * | 19.46 | 29.94 | 25.06 | 24.82 | 0.27 | 0.66 | 0.64 | 0.34 | 0.69 | 0.52 |
> | 0.6 | **TD-MoE**  | 20 | 8  | 0.60 | 12.40 | 19.88 | 20.96 | 17.75 | 0.27 | 0.73 | 0.68 | 0.41 | 0.75 | **0.568** |
>
>
> Using only 16 experts achieves **lossless performance at 20% compression** relative to the uncompressed model, indicating substantial redundancy across experts in Qwen2.5-57B. Specifically, we exclude the shared experts and apply tensorization only to the routed experts.

---

> ### Author Response · Authors · 2025-11-21
> **Response to Reviewer E6zg- Part #2**
>
> ### W2: Limited model coverage.
>
> A: Besides Mixtral-8×7B, we evaluated TD-MoE on Qwen2-57B-A14B (64 experts), Phi-3.5-MoE (K=16), and we provide the complete results below for clarity. The performance trend is consistent across both architectures. TD-MoE achieves the strongest perplexity and accuracy among decomposition baselines at 20–40% compression.
>
> Results on Qwen2-57B-A14B, please refer to [Response to Reviewer E6zg- Part #1](https://openreview.net/forum?id=D9cnZNZfxX&noteId=OvvNm2Uce5). Results on Phi-3.5-MoE are shown below.
>
> | Ratio | Method   | W2  | PTB  | C4  | Avg | OBQA | A_e | Wino | Hella | A_c | PIQA | MathQA | Avg |
> |-------|----------|-----|------|-----|--------|------|--------|--------|--------|--------|-------|--------|------|
> | origin | baseline | **3.5** | **8.4** | **8.2** | **6.7** | .40 | .77 | .76 | .68 | .56 | .79 | .38 | .62 |
> | 0.2 | ASVD      | 7.2 | 10.7 | 9.6 | 9.2 | .35 | .73 | .72 | .57 | .49 | .75 | .34 | .56 |
> |     | SVD-LLM   | 8.3 | 14.8 | 12.9 | 12.0 | .31 | .67 | .66 | .53 | .45 | .72 | .22 | .51 |
> |     | MoE-SVD   | **4.6** | 10.1 | 9.9 | 8.2 | .39 | .77 | .73 | .63 | .53 | .78 | .35 | .60 |
> |     | **Ours**  | 4.7 | **9.2** | **9.1** | **7.7** | **.39** | **.77** | **.74** | **.65** | **.55** | **.79** | **.38** | **.61** |
> | 0.4 | NAEE      | 8.2 | 20.1 | 16.1 | 14.8 | .35 | .73 | .73 | .61 | .48 | .76 | .37 | .57 |
> |     | MoE-I2    | 7.5 | 21.0 | 21.0 | 16.5 | .29 | .59 | .67 | .27 | .40 | .70 | .25 | .45 |
> |     | ASVD      | 14.5 | 22.1 | 21.8 | 19.5 | .30 | .69 | .63 | .41 | .40 | .68 | .25 | .48 |
> |     | SVD-LLM   | 38.8 | 68.5 | 43.8 | 50.4 | .23 | .56 | .60 | .39 | .31 | .66 | .24 | .43 |
> |     | MoE-SVD   | **5.5** | 11.7 | 11.9 | 9.7 | .35 | .72 | .72 | .58 | .48 | .75 | .31 | .56 |
> |     | **Ours**  | 6.4 | **10.9** | **10.8** | **9.4** | **.35** | **.75** | **.73** | **.61** | **.50** | **.78** | **.33** | **.58** |
> | 0.6 | ASVD      | 107.7 | 208 | 161 | 159 | .18 | .40 | .53 | .30 | .24 | .59 | .23 | .35 |
> |     | SVD-LLM   | 7168 | 7101 | 7119 | 7129 | .15 | .28 | .51 | .26 | .22 | .54 | .21 | .31 |
> |     | MoE-SVD   | **7.5** | 21.0 | 21.9 | **16.8** | .30 | .60 | .68 | .46 | .40 | .71 | **.25** | .49 |
> |     | **Ours**  | 13.7 | **19.7** | **17.9** | 17.1 | **.28** | **.70** | **.68** | **.46** | **.41** | **.73** | .23 | **.50** |
>
> *Phi-3.5-MoE with 16 experts.*

---

> ### Author Response · Authors · 2025-11-21
> **Response to Reviewer E6zg- Part #3**
>
> ### W3: Rank-allocation evaluation: quantitative comparisons.
> **A**: We thank the reviewer for highlighting the need for clearer explanation. BCRS, PRB, and FGAA are not independent heuristics; they form a progressive, hierarchical rank-allocation strategy.
> - BCRS (Budget-Constrained Rank Search) performs a simple rank search that matches the target compression budget, without considering model characteristics.
> - PRB builds on BCRS by adding an intra-tensor balancing term, preventing highly uneven allocations across modes and improving numerical stability.
> - FGAA operates at the model level, distributing per-layer budgets using Fisher-guided importance, with BCRS/PRB optionally applied as the second-stage optimizer within each layer.
>
> Finally, manual adjustments to specific layers (e.g., early layers) remain useful in practice. Together, these components form grouped strategy for rank allocation. We will add the detailed description in the paper.
>
> **Quantitative comparisons.**
> We first provide the layer-wise Fisher information of Mixtral-8x7B using Wikitext-2 below.
>
> | Layer | Fisher | Layer | Fisher | Layer | Fisher | Layer | Fisher |
> |------:|--------:|------:|--------:|------:|--------:|------:|--------:|
> | 0 | 1.64e−3 | 8 | 5.61e−5 | 16 | 6.27e−5 | 24 | 2.86e−5 |
> | 1 | 3.39e−3 | 9 | 1.91e−4 | 17 | 8.32e−5 | 25 | 1.00e−5 |
> | 2 | 1.41e−4 | 10 | 3.71e−5 | 18 | 3.58e−5 | 26 | 1.46e−5 |
> | 3 | 1.69e−4 | 11 | 1.04e−4 | 19 | 1.11e−4 | 27 | 8.98e−6 |
> | 4 | 1.21e−4 | 12 | 4.92e−5 | 20 | 3.43e−5 | 28 | 2.34e−4 |
> | 5 | 1.42e−4 | 13 | 1.30e−4 | 21 | 6.96e−6 | 29 | 6.06e−4 |
> | 6 | 9.20e−5 | 14 | 1.23e−4 | 22 | 2.36e−5 | 30 | 4.42e−4 |
> | 7 | 8.13e−5 | 15 | 8.75e−5 | 23 | 1.50e−5 | 31 | 1.35e−3 |
>
> Below shows the summary of the statistical properties of the allocated Tucker ranks under different strategies.
>
> | Ratio | Method | $r_{\text{out}}$ (mean / range) | $r_{\text{in}}$ (mean / range) | Balance (mean $\vert r_{\text{out}}/r_{\text{in}} \vert$) |
> | :---: | :--- | :--- | :--- | :---: |
> | **0.2** | Uniform | 2461 / (2461–2461) | 8614 / (8614–8614) | 0.29 |
> | | PRB-only | 3661 / (3661–3661) | 8271 / (8271–8271) | 0.44 |
> | | FGAA-only | 2963 / (1360–4096) | 8412 / (3072–13403) | 0.33 |
> | | PRB+FGAA | 2404 / (914–3988) | 9604 / (3199–13959) | 0.25 |
> | **0.4** | Uniform | 2241 / (2241–2241) | 7845 / (7845–7845) | 0.29 |
> | | PRB-only | 2585 / (2585–2585) | 7747 / (7747–7747) | 0.33 |
> | | FGAA-only | 2413 / (874–4096) | 8247 / (2620–13733) | 0.28 |
> | | PRB+FGAA | 2185 / (758–3973) | 8891 / (2654–13907) | 0.24 |
>
> Below shows the downstream results.
> | Ratio | Method | Avg Acc  |
> |:----:|:--------|:--------------:|
> |0.2| Uniform |  0.63 |
> |0.2| PRB-only |   0.63|
> |0.2| FGAA-only |   0.64 |
> |0.2| PRB+FGAA |  0.65  |
> |0.4| Uniform |    0.60|
> |0.4| PRB-only |   0.60 |
> |0.4| FGAA-only |   0.59 |
> |0.4| PRB+FGAA |   0.60 |

---

> ### Author Response · Authors · 2025-11-21
> **Response to Reviewer E6zg- Part #4**
>
> ### W4: Inference-time analysis.
>
> A: We clarify how Tucker-based reconstruction affects inference-time complexity, and how our triple-GEMM kernel resolves the main bottleneck.
>
> **(i) Where the slowdown comes from.**
> After Tucker decomposition and absorbing the mode-0 factor, each expert weight can be written as $W_e \approx U_1 \, G_e \, U_2^\top$, and the expert forward becomes $Y_e = X \, U_2 \, G_e \, U_1^\top$.
>
> Mathematically, this replaces the original dense GEMM $XW_e$ (1 GEMM) by a three-matrix chain $X \, U_2 \, G_e \, U_1^\top$. In a naïve implementation, this is evaluated as **three separate GEMMs** with two intermediate matrices written to and read from global memory. Even though the Tucker ranks reduce the FLOPs compared with the original dense GEMM, profiling on Phi-3.5-MoE shows that about 70% of the time is spent in GEMMs and ≈30% in kernel launches and intermediate materialization. This explains the observed throughput degradation:
>
> | Compression ratio | Throughput (token/s) | Relative to baseline | Per-token time |
> |------------------|----------------------|----------------------|-----------------|
> | **0% (dense)**   | **569**              | 1.00×                | 1.00× |
> | **60%**          | **485**              | 0.85×                | 1.17× |
> | **20%**          | **318**              | 0.56×                | 1.70× |
>
> *Phi-3.5-MoE, 2×A800, bf16, bs=32, seqlen=32, Wikitext-2 generation.*
>
> In other words, Tucker already reduces arithmetic cost, but the extra GEMM launches and DRAM traffic dominate, so end-to-end latency becomes worse than the dense baseline.
> **(ii) How the triple-GEMM kernel fixes this at the operator level.**
> The key observation is that, after absorbing one factor offline, the Tucker-reconstructed expert matmul can always be written in the three-matrix form: $Y = X U_2 (G_e U_1^\top)$, where $A$ and $B$ are products of $U_1$, $U_2$, $G_e$. A standard implementation evaluates this as two back-to-back GEMMs. Our fused triple-GEMM kernel computes the same expression **in a single kernel launch**. The intermediate tiles are kept entirely on-chip (registers / shared memory) and forwarded between stages, eliminating both the extra global-memory round trip and the extra kernel launch. The measured TFLOPS numbers against cuBLAS back-to-back GEMMs for the same three-matrix chain are:
>
> | $d_{\text{in}}$ (Input Dim) | $d_{\text{out}}$ (Output Dim) | $r$ (Tucker Rank) | $p$ (Tokens) | Baseline (TFLOPS) | Ours (TFLOPS) | Speedup |
> |:---------------------------:|:------------------------------:|:-----------------:|:------------:|:------------------:|:--------------:|:--------:|
> | 4096, 11008                 | 4096, 11008                    | 640–2400          | 512–8192     | 20.78              | **32.14**      | **1.58×** |
> | 5120, 13824                 | 5120, 13824                    | 640–2976          | 512–8192     | 18.61              | **31.82**      | **1.87×** |
> | 1024, 8192, 28672           | 5120, 13824                    | 90–2457           | 512–8192     | 12.33              | **28.41**      | **3.07×** |
>
> This shows that, for the same Tucker ranks, the triple-GEMM fused kernel makes the $Y=XAB$ part itself up to 1.6–3.1× faster by removing launch and DRAM overhead.
>
> **(iii) Speedup Estimation**
> Since integrating the fused triple-GEMM kernel into the MoE pipeline requires non-trivial modifications, the engineering effort exceeds what can be completed within the rebuttal window. We therefore provide a speedup analysis.
> - Let the current compressed MoE runtime be $T_\text{comp}$, with fraction  $p \approx 0.7$ of time spent inside the triple GEMMs.
> - Replacing the three cuBLAS GEMMs by our fused kernel makes this GEMM portion ~3× faster. By Amdahl’s law, the new runtime is $
>   T_\text{new} \approx T_\text{comp} \bigl( (1-p) + \tfrac{p}{3} \bigr)
>   \approx T_\text{comp} \times 0.53.
>   $
>
> Using Amdahl’s law, we obtain the end-to-end estimates already reported:
>
> - For 20% compression, $T_\text{comp} \approx 1.79 T_\text{base}$ ⇒   $T_\text{new} \approx 0.95 T_\text{base}$, corresponding to  ~1.05× baseline throughput.
> - For 60% compression, $T_\text{comp} \approx 1.17 T_\text{base}$ ⇒   $T_\text{new} \approx 0.62 T_\text{base}$, corresponding to  ~1.6× baseline throughput.
>
> These estimates are approximate but realistic. With a production-quality fused triple-GEMM kernel, a Tucker-compressed MoE can match—or even exceed—the throughput of the original MoE while still achieving 20–60% parameter reduction.

---

> ### Author Response · Authors · 2025-11-21
> **Response to Reviewer E6zg- Part #5**
>
> ### W5: Combination with quantization/pruning left untested.
>
> **A:** To verify that TD-MoE is compatible with post-training methods:
> **Quantization.** we added a combination study using 8-bit NF4 (bitsandbytes) applied after TD-MoE compression on Mixtral-8×7B. We compare the original model, TD-MoE (20%/40%), and TD-MoE + 8bit variants.
>
> | Ratio | Wiki  | PTB  | C4  |  Avg ppl | OBQA | ARC-E | WinoG | Hella | ARC-C | PIQA | MathQA | Avg Acc|
> |:-----:|:------:|:------:|:-----:|:---------:|:-----:|:------:|:-------:|:-------:|:-------:|:------:|:--------:|:---------------:|
> | Origin | 3.98 | 12.99 | 6.78 | 7.92 | 0.36 | 0.84 | 0.76 | 0.65 | 0.57 | 0.82 | 0.43 | 0.63 |
> | 0.2 | 4.49 | 17.20 | 7.73 | 9.81 | 0.33 | 0.83 | 0.77 | 0.64 | 0.53 | 0.82 | 0.40 | 0.62 |
> |0.2 + 8-bit | 4.50 | 16.87 | 7.74 | 9.70 | 0.33 | 0.83 | 0.77 | 0.64 | 0.53 | 0.82 | 0.40 | 0.62 |
> | 0.4 | 5.79 | 24.60 | 9.21 | 13.20 | 0.28 | 0.77 | 0.76 | 0.57 | 0.47 | 0.79 | 0.35 | 0.57 |
> | 0.4 + 8-bit | 5.85 | 24.47 | 9.26 | 13.20 | 0.28 | 0.77 | 0.73 | 0.57 | 0.45 | 0.79 | 0.35 | 0.57 |
>
> Applying 8-bit quantization on top of TD-MoE introduces negligible additional degradation, with LM average changes ≤ 0.1, and reasoning accuracy changes ≤ 0.01.
> **Pruning.** We perform structured pruning directly in the Tucker space by ranking each mode-1 and mode-2 component according to the ℓ2-energy of their corresponding slices in the core tensor. Low-energy rank components are removed together with their associated factor-matrix columns, yielding a lower-rank yet structurally valid Tucker decomposition.
>
> | Comp. Ratio | Sparsity | Wikitext-2  | PTB  | Avg  | OBQA  | A_E  | Wino  | A-C  | PIQA  | Avg  |
> |--------|----------|---------------|--------|--------|---------|----------|----------|----------|---------|----------------|
> | 20%  | 0%  | 4.49 | 17.20 | 10.85 | 0.33 | 0.83 | 0.77 | 0.53 | 0.82 | 0.66 |
> | 20%  | 20% | 4.99 | 20.44 | 12.72 | 0.31 | 0.81 | 0.77 | 0.51 | 0.79 | 0.64 |
> | 20% | 30% | 5.35 | 23.07 | 14.21 | 0.30 | 0.79 | 0.76 | 0.48 | 0.79 | 0.62 |
> | 20% | 40% | 5.80 | 25.64 | 15.72 | 0.28 | 0.78 | 0.75 | 0.46 | 0.79 | 0.61 |
> | 40% | 0%  | 5.79 | 24.60 | 15.20 | 0.28 | 0.77 | 0.76 | 0.47 | 0.79 | 0.57 |
> | 40% | 40% | 10.20 | 49.92 | 30.06 | 0.23 | 0.67 | 0.70 | 0.35 | 0.74 | 0.54 |
>
> As shown above, increasing the pruning ratio progressively reduces parameter count and yields a controlled degradation in accuracy. This demonstrates that pruning and tensor decomposition are complementary.
>
> ### W6: Minor writing/consistency issues.
> A: Thank you for your comments. We will carefully revise the terminology, correct typos, and ensure full consistency in the camera-ready version.

---

### Official Review · Reviewer_hRUS · 2025-10-30

**Soundness:** 4
**Presentation:** 3
**Contribution:** 3
**Rating:** 6
**Confidence:** 4

**Summary:**

The paper presents a principled approach to MoE compression by reformulating it as joint tensor decomposition, effectively capturing cross-expert redundancy. The core innovation of stacking experts into a 3D tensor and performing Tucker decomposition captures shared structure beyond the per-expert scope (Sec. 3.1; Fig. 2; p.4). Strong experimental results show TD-MoE achieves 0.57 accuracy vs. 0.50 (MoE-SVD) at 40% compression on Mixtral-8×7B commonsense tasks and maintains low perplexity (4.49/17.20/7.73) at 20% compression (Tables 1-2; Sec. 4.2; p.8). However, the computational overhead and storage requirements for the covariance matrices (Σin, Σout) used in multi-linear whitening are not fully quantified (Sec. 3.3; p.5). Implementation details regarding hyperparameter selection also need clarification.

**Strengths:**

- **Novel cross-expert tensorization with strong theoretical foundation**
  - The method unifies K experts into a tensor $\mathcal{T} \in \mathbb{R}^{K \times d_{\text {in }} \times d_{\text {out }}}$ and applies Tucker decomposition $\mathcal{T} \approx \mathcal{G} \times_1 \mathbf{U}^{(1)} \times_2 \mathbf{U}^{(2)} \times_3 \mathbf{U}^{(3)}$, where the factors have clear interpretations: U1 defines meta-experts, and U2/U3 span low-dimensional input/output subspaces (Eq. 3; Sec. 3.2; p.5), providing a principled factorization.
  - Cross-expert factorization addresses the limitations of per-expert methods that overlook structural redundancies across experts trained on related distributions (Sec. 2.1; p.3), improving compression efficiency.
  - Tucker decomposition naturally compresses the expert dimension via the U1 factor without requiring explicit expert dropping/merging decisions (Sec. 3.2; p.5), offering graceful, data-driven redundancy reduction.
- **Effective multi-linear whitening strategy**
  - Multi-linear whitening decorrelates input/output modes via $S_{\text{in}} = (\Sigma_{\text{in}} + \epsilon I)^{-1/2}$ and $S_{\text{out}} = (\Sigma_{\text{out}} + \epsilon I)^{-1/2}$, producing a well-conditioned tensor $\mathcal{T}\_w = \mathcal{T} \times\_2 S\_{\text{in}} \times\_3 S\_{\text{out}}$ (Eq. 4; Sec. 3.3; p.6), improving decomposition stability.
  - The method adapts to the statistical geometry of the data by collecting both input activations X and output gradients ∇YL, capturing correlations and sensitivities (Sec. 3.3; p.5), yielding a data-adaptive factorization.
  - Re-coloring for inference absorbs the inverse whitening into the factors $U^{\prime(1)}=U_1, U^{\prime(2)}=S_{i n}^{-1} U_2$ and $U^{\prime(3)}=S_{\text {out }}^{-1} U_3$, ensuring no runtime overhead (Sec. 3.3; Fig. 4b; p.9), maintaining efficiency.
- **Comprehensive experimental validation with strong baselines**
  - At 20% compression on Mixtral-8×7B, output-whitening achieves 0.62 average accuracy vs. 0.63 for the uncompressed model, a <1% loss across seven benchmarks (Table 1; Sec. 4.2; p.8), demonstrating near-lossless compression.
  - Language modeling results show 4.49/17.20/7.73 perplexity (WikiText-2/PTB/C4) at 20% compression, significantly outperforming NAEE and MoE-SVD (Table 2; Sec. 4.2; p.8), validating effectiveness.
  - Ablations systematically examine whitening directions (input vs. output vs. both) and Tucker backends (QR vs. rand_svd vs. SVD), confirming design choices (Fig. 4; Sec. 4.3; p.9), supporting technical soundness.

**Weaknesses:**

- **Limited analysis of computational and memory overhead**
  - Whitening requires computing and storing $\Sigma_{\text{in}} = \frac{1}{N} X^T X, \quad \Sigma_{\text{out}} = \frac{1}{N} (\nabla_Y \mathcal{L})^T (\nabla_Y \mathcal{L})$ per layer (Sec. 3.3; p.5). For large models ($d_in, d_out~10^4$), this incurs $O(d^2)$ memory and $O(Nd^2)$ computation, but overhead quantification is absent.
  - Cholesky decomposition for whitening matrices has $O(d^3)$ complexity (Sec. 3.3; Algorithm 1; p.13). Scalability to very large dimensions is not discussed. (Note: Algorithm 1 is in Appendix A.1).
  - Tucker decomposition computational complexity is not analyzed relative to per-expert SVD baselines (Sec. 3.2; p.5). Practical runtime comparisons beyond offline compression time are missing.
- **Hyperparameter selection and rank allocation details**
  - The BCRS method fixes $r_0=K$ and searches ($r_1,r_2$) to satisfy the budget constraint (Eq. 5; Sec. 3.4; p.6), but the rationale for fixing the expert rank is not explained—allowing r0<K could improve compression.
  - Proportional Rank Balancing and Fisher-Guided Adaptive Allocation are mentioned, but details are relegated to the appendix (Sec. 3.4; Appendix A.3; p.6,13). The main text lacks sufficient explanation of rank selection strategies.
  - The covariance eigenvalue clipping threshold ($10^{-3}$) is specified but not justified (Sec. 4.1; p.7). Sensitivity to this hyperparameter is unexplored.
- **Mathematical notation and presentation issues**
  - Tucker decomposition notation $\mathcal{T} \approx \mathcal{G} \times_1 \mathbf{U}^{(1)} \times_2 \mathbf{U}^{(2)} \times_3 \mathbf{U}^{(3)}$ uses the mode-n product (×n) without defining it in the main text (Eq. 3; Sec. 3.2; p.5). Readers unfamiliar with tensor algebra may struggle.
  - The compression ratio formula $P_{orig}= Kd_{out}d_{in}$ and $P_{tucker}=r_0r_1r_2+Kr_0+d_{out}r_1+d_{in}r_2$ appears without derivation (Sec. 3.4; Appendix A.3; p.13). A step-by-step derivation would improve clarity.
  - Figure 2 shows the overall pipeline but lacks a detailed illustration of how the whitened tensor Tw differs from the original T (Fig. 2; p.4). Additional visualizations would aid understanding.
- **Limited scope of experimental evaluation**
  - Experiments focus on Mixtral-8×7B and Phi-3.5-MoE; evaluation on larger models (e.g., models with >100B parameters or >16 experts) would strengthen generalizability claims (Sec. 4.1; p.7).
  - Only 512 calibration samples were used for whitening statistics (Sec. 4.1; p.7). A sensitivity analysis varying the calibration set size (e.g., 128-2048 samples) is absent.
  - Language modeling was evaluated only on WikiText-2, PTB, C4 (Table 2; p.8). Additional domains (code, multilingual) would demonstrate broader applicability.

**Questions:**

- **Quantify computational and memory overhead**
  - Provide a detailed complexity analysis comparing TD-MoE vs. per-expert SVD: decompose the total cost into calibration ($O(Nd^2)$), whitening (O(d^3)), and Tucker decomposition, and provide wall-clock timing for each stage across model scales.
  - Report memory footprints explicitly: covariance matrix storage ($2d^2$), whitening matrix storage ($2d^2$), Tucker factors storage ($r_0r_1r_2+Kr_0+d_{out}r_1+d_{in}r_2$), and peak memory during decomposition. Include scalability plots showing overhead vs. ($d_{in}, d_{out}$).
  - Discuss optimization strategies for very large dimensions, such as incremental covariance updates, randomized whitening approximations, or block-diagonal approximations with empirical validation.
- **Clarify rank allocation methodology**
  - Justify fixing $r_0=K$ in BCRS: either provide a theoretical rationale or conduct ablations varying $r_0 \in \left[\frac{K}{4}\, K\right]$ to show the impact on compression-accuracy tradeoffs. Report results in the main text with supplementary details in the appendix.
  - Move the descriptions of Proportional Rank Balancing and Fisher-Guided Adaptive Allocation from the appendix to the main text with concrete algorithms and complexity analysis (currently Appendix A.3; integrate into Sec. 3.4).
  - Perform a sensitivity analysis on the eigenvalue clipping threshold: vary $\varepsilon \in \{10^{-4}, 10^{-3}, 10^{-2}, 10^{-1}\}$
 and report perplexity/accuracy impacts with recommendations for different model scales.
- **Improve mathematical presentation**
  - Define the mode-n product (×n) explicitly in the main text with a small worked example (e.g., a 2×2×2 tensor) before introducing the full Tucker decomposition. Include a visual diagram showing how factors combine to reconstruct the tensor.
  - Provide a step-by-step derivation of the parameter counts: start from T∈R^{K×dout×din}, apply the Tucker definition, count the free parameters in G and each factor, and show how the compression ratio formula arises (currently implicit in Sec. 3.4; p.6).
  - Add a whitening visualization: show heatmaps of Σin before/after whitening, singular value distributions of T vs. Tw, and reconstruction error distributions to illustrate the benefits of whitening (extending Fig. 2; p.4).
- **Expand experimental scope**
  - Evaluate on larger-scale models: include experiments on models with 100B+ parameters or 32+ experts per layer to demonstrate scalability. Report compression ratios, perplexity, and downstream accuracy.
  - Conduct a calibration set size sensitivity analysis: vary N∈{128, 256, 512, 1024, 2048} and measure whitening matrix stability (e.g., condition number convergence) and the resulting model performance.
  - Broaden task diversity: add code generation (HumanEval), multilingual tasks (XNLI/XQuAD), and domain-specific benchmarks (scientific QA) to demonstrate cross-domain effectiveness beyond commonsense reasoning and English language modeling.

---

> ### Author Response · Authors · 2025-11-21
> **Response to Reviewer hRUS- Part #1**
>
> We sincerely thank the reviewer for the insightful and constructive feedback. Your comments greatly improved the clarity and rigor of our work. Below, we address each of your concerns in detail.
>
> ### W1: Limited analysis of computational and memory overhead
> Thank you for your comments. We have added the detailed computational and memory overhead of whitening, cholesky decomposition and Tucker decomposition.
>
> >**W1-Q1**: Computational and storage overhead of whitening
>
> **A**: Whitening process consumes less than 1% extra memory and a small computational overhead of 0.11 TFLOPs. The required memory of input and output whitening is shown below.
>
>
> | Model  |  Memory Overhead| % compared to the full model size (FP16) |
> |--------------------------------------|-------------------|----------------------------|
> | Mixtral-8×7B (46B)  | **890MB** | **≈1%** |
> | Phi-3.5-MoE (42B)   | **231 MB** | **≈0.4%** |
>
> (*input whitening*)
>
> | Model / FFN dims | $d_{\text{out}}$ | Grad Buffer | Covariance Cost | Grad Buffer (Streaming, 32-block) |
> |------------------|--------------------|-------------|------------------|-----------------------------------|
> | Mixtral-8×7B     | 14336              | 14 MB   | 0.11 TFLOPs  | **0.88 MB**                       |
> | Phi-3.5-MoE      | 6400               | 6.3 MB  | 0.02 TFLOPs  | **0.39 MB**
>
> (*output whitening*)
>
> It's worth noting that whitening is a one-time offline preprocessing step applied layer-by-layer. All intermediate tensors are freed immediately after each layer is processed and therefore never appear in the runtime graph. And the whitening is applied to the input/output modes of the stacked expert tensor once, and no need to apply it to each expert individually.
>
> > **W1-Q2**: Scalability to very large dimensions of Cholesky decomposition.
>
> **A**: In our setting,  Cholesky decomposition can scale to very large dimensions up to 14336, and is not a bottleneck in practice. Below shows the measured cost of decomposing dense matrices using A800 varying matrix sizes.
>
> | Size $d$ | Cholesky Time (ms) | Memory (fp32) |
> |------------|--------------------|----------------|
> | 2048 | 1.66 | 16 MB |
> | 3072 | 2.61 | 36 MB |
> | 4096 | 4.13 | 64 MB |
> | 5120 | 7.07 | 100 MB |
> | 6400 | 11.44 | 156 MB |
> | 7168 | 14.44 | 196 MB |
> | 8192 | 20.03 | 256 MB |
> | 14336 | 77.43 | 784 MB |
>
> For Mixtral-8×7B (32 MoE layers), output-only whitening requires 32*77.43ms≈2.5s, and input+output whitening remains around 2.6s. The peak extra memory stays below 1 GB and does not scale with $K$. Since decomposition is performed layer-wise, memory is freed immediately after each layer, and the preprocessing can be further accelerated through layer-wise parallelism (e.g., CUDA streams) or low-rank/randomized whitening.
>
> **Analysis.** Decomposition is applied only to 2D covariance matrices, not to the 3D expert tensor. Each MoE layer requires at most one Cholesky per direction (input/output). This preprocessing is done once per layer and is independent of the Tucker ranks or the number of experts $K$, unlike SVDLLM or MoE-SVD, whose cholesky decomposition number is scaling with the number of experts $K$.

---

> ### Author Response · Authors · 2025-11-21
> **Response to Reviewer hRUS- Part #2**
>
> > **W1-Q3**: Tucker complexity v.s. per-expert SVD
>
> **A:** Tucker has the asymptotic computational complexity with per-expert SVD, is computationally competitive or even faster than per-expert SVD.
>
> **Complexity Analysis.** For the per-expert SVD baseline, the computational cost is $E \cdot O\!\bigl(MN \min(M,N)\bigr)$. Tucker decomposition performs three SVDs over the mode-\(k\) unfoldings, giving cost as $O(EMN) + O(MEN) + O(NEM)$. Using randomized SVD reduces each SVD to $O(MNr)$, where $r$ is the target rank. Because the dominant unfolding is the mode-1 unfolding (approximately an $M \times EN $matrix), randomized SVD provides substantial efficiency improvements in practice.
>
> **Empirical Runtime (Offline).** We further compared the runtime of Tucker decomposition v.s. per-expert SVD on two MoE models with varying expert dimension and number.
>
> | Compression | Tucker (rand) | 8× SVD  | Relative Speed |
> |------------|---------------|---------------|----------------|
> |0.2 | 20.7 s | 9.42 s | 2.2× slower |
> | 0.4 | 12.5 s | 9.40 s |  1.3× slower |
> | 0.6 | 7.52 s | 9.39 s | **1.25× faster** |
>
> *Mixtral-8×7B (E=8, M=14336, N=4096)*
>
> As the reduction ratio increases (i.e., we remove more parameters and ranks become smaller), Tucker becomes **faster** than the per-expert baseline. At compression ratio = 0.4, the gap is small (~3 s) and all costs are offline.
>
> For Phi-3.5-MoE, Tucker is consistently much faster due to more experts and smaller per-expert matrices.
>
> | Compression | Tucker (rand) | 16× SVD (full) | Relative Speed |
> |------------|---------------|----------------|----------------|
> | 0.2 | 7.07 s | 17.71 s | **2.5× faster** |
> | 0.4 | 5.56 s | 17.68 s | **3.2× faster** |
> | 0.6 | 3.88 s | 17.66 s | **4.6× faster** |
>
> *Phi-3.5-MoE (E=16, M=6400, N=4096)*
> Tucker is consistently 2.5×–4.6× faster across all reduction ratios due to the larger number of experts and smaller per-expert matrices, which favor mode-1 randomized SVD. Across both MoE models, the results demonstrate that Tucker decomposition is computationally efficient, scalable, and never introduces a practical bottleneck relative to per-expert SVD.
>
> ### W2 Hyperparameter selection and rank allocation details
>
> > **W2-Q1**: The rationale for fixing the expert rank is not explained.
>
> A:  Our method supports and is effective for varying $K$. For extra experimental results, please refer to [Response to Reviewer hRUS- Part #6](https://openreview.net/forum?id=D9cnZNZfxX&noteId=FDxYo1UTZz). Our method obtains lossless at 20% compression ratio in compressing Qwen2-57B-A14B, compressing K from 64 to 16/8.
>
> The rationale for fixing K was to ensure a fair comparison with the decomposition-based MoE-SVD baseline because reducing the number of experts in MoE-SVD leads to significant performance degradation. Therefore, we fixed $K$ to isolate the benefits of our method. Still, our method provides a tensorization-based joint compression method for MoE, supporting searching across all three tensor ranks.
>
> > **W2-Q2**: The main text lacks sufficient explanation of rank selection strategies.
>
> **A**: Thank you for your comments. As the main focus of the paper is on tensorization-based decomposition and multi-linear whitening, the discussion of rank selection strategies in the main text is intentionally kept concise. Nevertheless, we agree that further clarification would improve readability, and we will expand the explanation of our rank selection principles in the main paper for better completeness.

---

> ### Author Response · Authors · 2025-11-21
> **Response to Reviewer hRUS- Part #3**
>
> > **W2-Q3**: The covariance eigenvalue clipping threshold.
>
> **A:** We included a sensitivity experiment varying clipping threshold $\epsilon \in \{10^{-1},10^{-2},10^{-3},10^{-4}\}$ under two compression ratios (20% and 40%) on Mixtral-8×7B. The results show that $\epsilon$ has limited effect on performance, with variations limited to only 0.01–0.02 in perplexity and ≤0.02 on downstream accuracy.
>
> | Compression Ratio | Threshold | WikiText2 | PTB | Avg PPL | OpenBook | ARC-e | WinoG | ARC-c | PIQA | Avg Acc|
> |-------|-----------|-----------|-----|----------|-----------|--------|--------|---------|--------|-------------------|
> | origin | — | 3.98 | 12.99 | 7.92 | 0.36 | 0.84 | 0.76 | 0.57 | 0.82 | 0.63 |
> | **0.2** | 1e-1 | 4.50 | 17.46 | 7.32 | 0.32 | 0.82 | 0.76 | 0.53 | 0.81 | 0.65 |
> |        | 1e-2 | 4.50 | 17.46 | 7.32 | 0.32 | 0.82 | 0.76 | 0.53 | 0.81 | 0.65 |
> |        | 1e-3 | **4.49** | **17.20** | **7.23** | **0.33** | **0.83** | **0.77** | **0.53** | **0.82** | **0.66** |
> |        | 1e-4 | 4.50 | 17.46 | 7.32 | 0.32 | 0.82 | 0.76 | 0.53 | 0.81 | 0.65 |
> | **0.4** | 1e-1 | 5.80 | 24.80 | 10.20 | 0.28 | 0.77 | 0.75 | 0.47 | 0.78 | 0.61 |
> |        | 1e-2 | 5.80 | 24.80 | 10.20 | 0.28 | 0.77 | 0.75 | 0.47 | 0.78 | 0.61 |
> |        | 1e-3 | **5.79** | **24.60** | **10.13** | 0.28 | 0.77 | **0.76** | 0.47 | **0.79** | **0.61** |
> |        | 1e-4 | 5.80 | 24.80 | 10.20 | 0.28 | 0.77 | 0.75 | 0.47 | 0.78 | 0.61 |
>
> Across all eight downstream tasks and two language-modeling benchmarks, the curves are effectively flat with respect to $\epsilon$. This is expected: covariance whitening primarily removes extremely small eigenvalues arising from numerical noise, and such components contribute negligibly to the decomposition. We will integrate this clarification and the sensitivity table into the main text.
>
> ### W3 Mathematical notation and presentation issues
> > **W3-Q1**: Lack of clear notation of Tucker decomposition, such as mode-n product.
>
> **A**:  In the revised version, we will explicitly define the mode-$n$ tensor–matrix product when it first appears in Sec. 3.2 and provide the element-wise Tucker reconstruction notations $\mathcal{T}_{i_1 i_2 i_3}\approx\sum _{r_1,r_2,r_3} \mathcal{G} _{r_1 r_2 r_3} U^{(1)} _{i_1 r_1} U^{(2)} _{i_2 r_2}U^{(3)} _{i_3 r_3}$. This expansion makes the notation self-contained for readers unfamiliar with tensor algebra.
>
> > **W3-Q2**: Lack of a step-by-step derivation of compression ratio formula.
>
> **A**: We provide detailed formulatation about the compression ratio derivation. Given the original MoE FFN weights tensorized as $\mathcal{T}\in\mathbb{R}^{K\times d_{\text{out}}\times d_{\text{in}}}$, which contains $P_{\text{orig}} = K\, d_{\text{out}} d_{\text{in}}$ parameters; Under the Tucker decomposition with core $\mathcal{G}\in\mathbb{R}^{r_0\times r_1\times r_2}$ and factor matrices $U^{(0)}\in\mathbb{R}^{K\times r_0}$, $U^{(1)}\in\mathbb{R}^{d_{\text{out}}\times r_1}$,  $U^{(2)}\in\mathbb{R}^{d_{\text{in}}\times r_2}$, the total number of parameters becomes $P_{\text{tucker}} ={r_0 r_1 r_2} (Core) + {(K r_0 + d_{\text{out}} r_1 + d_{\text{in}} r_2)} (Factor Matrices).$ We will add this into the main paper for clear clarification.
>
> > **W3-Q3**: Lack of clear illustration of the whitened tensor in Figure 2.
>
> **A**: Thank you for the suggestion. The current figure shows the high-level workflow of our tensorization and whitening pipeline, but we agree that the whitened tensor itself could be illustrated more clearly. In the revision, we will refine Figure 2 by adding intermediate visual cues (e.g., tensor shapes and whitening transformations) to make the whitening effect more explicit.

---

> ### Author Response · Authors · 2025-11-21
> **Response to Reviewer hRUS- Part #4**
>
> ### Q1 Quantify computational and memory overhead
>
> > **Q1-Q1**: Complexity analysis comparing TD-MoE vs. per-expert SVD
>
> **A:** Following the method pipeline:
> -(1) Calibration cost. The calibration overhead of TD-MoE is identical to that of per-expert SVD, with complexity of $O(N d_{\text{in}}^2 + N d_{\text{out}}^2)$. For Mixtral-8×7B, this amounts to roughly 0.11 TFLOPs when using 512 calibration samples—less than 5% of the total preprocessing cost—and is incurred only once offline.
> (2) Whitening cost. Please refer to [W1-Q1](https://openreview.net/forum?id=D9cnZNZfxX&noteId=n9r2CESGS9) for the detailed whitening analysis.
> (3) Decomposition cost. The full decomposition complexity is provided in [W1-Q3](https://openreview.net/forum?id=D9cnZNZfxX&noteId=5FMPBu3LHM).
>
> >**Q1-Q2**: Memory footprints of matrices and peak memory during decomposition.
>
> **A**: We report memory footprints for covariance/whitening, Tucker factors, per-expert SVD factors, and peak decomposition memory. Results (fp32, $K=8$) across multiple $(d_{\text{in}}, d_{\text{out}})$ settings are shown below.
>
> | $d_{\text{in}}$ | $d_{\text{out}}$ | $Cov/Whiten$ | Tucker Total | Per-expert SVD | Peak Tucker | Peak SVD |
> |------------------:|-------------------:|------------:|--------------:|-------------:|-------------:|----------:|
> | 2048 | 4096  | 64 MB  | 0.19 GB | 0.19 GB | 0.44 GB | 0.44 GB |
> | 4096 | 6400  | 0.15 GB | **0.24 GB** | **0.32 GB** | 1.02 GB | 1.10 GB |
> | 4096 | 14336 | 0.77 GB | **0.30 GB** | **0.56 GB** | 2.05 GB | 2.31 GB |
> | 6400 | 14336 | 0.77 GB | **0.33 GB**| **0.63 GB** | 3.07 GB | 3.37 GB |
> | 8192 | 16384 | 1.00 GB | **0.38 GB** | **0.75 GB** | 4.38 GB | 4.75 GB |
>
> Tucker remains more memory-efficient and scales better than per-expert SVD acrossall tested FFN sizes, even on extremely large dimension [8192,16384]. This advantage comes from two sources:(1) Tucker avoids storing K separate SVD factorizations, replacing them with a shared core and mode matrices, and (2) the covariance and whitening operations are performed in reduced-dimensional subspaces, preventing quadratic growth in intermediate activations. A scalability plot will be included in the appendix of the reivsed paper.
>
>
> ### Q2 Clarify rank allocation methodology
> >**Q2-Q1**: Justifying fixing K in BCRS.
>
> **A**: Please refer to [W2-Q1](https://openreview.net/forum?id=D9cnZNZfxX&noteId=5FMPBu3LHM) for our justification and detailed explanation. The experiments for varying K, please refer to [Response to Reviewer hRUS- Part #6](https://openreview.net/forum?id=D9cnZNZfxX&noteId=FDxYo1UTZz).
>
> >**Q2-Q2**:  Descriptions of Proportional Rank Balancing and Fisher-Guided Adaptive Allocation.
>
> **A**: We will reorganize the rank-allocation section and integrate the descriptions of both strategies into the main text to improve clarity. Thank you for the suggestion.
>
> >**Q2-Q3**: Sensitivity analysis on the eigenvalue clipping threshold.
>
> **A**: Model perplexity and accuracy remain highly robust with respect to the eigenvalue-clipping threshold. Please refer to the detailed statistics provided in [W2-Q3](https://openreview.net/forum?id=D9cnZNZfxX&noteId=O2xVmnxfQa).
>
>
> ### Q3 Improve mathematical presentation
> >**Q3-Q1**: Formulation of mode-n operation & visualization of factor combination for reconstruction.
>
> **A**: Thank you for your suggestion. We will add the mode-n operation formulation in the paper, with an concise example like 2×2×2 tensor mentioned and clear visual diagram for tensor reconstruction demonstration.
>
> >**Q3-Q2**: Provide step-by-step derivation of the parameter counts.
>
> **A**: We agree and will make the parameter-count derivation explicit in the revised version. Concretely, we start from the stacked MoE tensor $\mathcal{T}\in\mathbb{R}^{K\times d_{\text{out}}\times d_{\text{in}}}$, which contains $P_{\text{orig}} = K\,d_{\text{out}}d_{\text{in}}$ parameters. Under a Tucker model $\mathcal{T} \approx \mathcal{G}\times_0 U^{(0)} \times_1 U^{(1)} \times_2 U^{(2)}$ with core $\mathcal{G}\in\mathbb{R}^{r_0\times r_1\times r_2}$ and factor matrices $U^{(0)}\in\mathbb{R}^{K\times r_0}$, $U^{(1)}\in\mathbb{R}^{d_{\text{out}}\times r_1}$, $U^{(2)}\in\mathbb{R}^{d_{\text{in}}\times r_2}$, the core contributes $r_0 r_1 r_2$ parameters and the factors contribute $K r_0 + d_{\text{out}} r_1 + d_{\text{in}} r_2$. Thus the total number of parameters after compression is $P_{\text{tucker}} = r_0 r_1 r_2 + K r_0 + d_{\text{out}} r_1 + d_{\text{in}} r_2$, and the compression ratio used in Sec. 3.4 becomes $\text{CR} = P_{\text{tucker}} / P_{\text{orig}}$. We will add this step-by-step derivation to Sec. 3.4 for clarity.

---

> ### Author Response · Authors · 2025-11-21
> **Response to Reviewer hRUS- Part #5**
>
> >**Q3-Q3**: Add a whitening visualization: Heatmaps of Σin before/after whitening,  singular value distributions of T vs. Tw, and reconstruction error distributions.
>
> **A**: We will add the whitening visualization for clearer clarification. Below shows the key statistics for visualization.
>
> We conduct the experiments to check the (i) before/after covariance matrices, (ii) eigenvalue distribution and (iii) correlation values. Specifically, we use the statistics of gate matrices on six layers (3, 5, 7, 9, 12, 24) using Mixtral-8×7B. All results demonstrate numerical exactness and strong cross-layer stability.
>
> **(1) Become Near Identical Matrices After Whitening.** For every layer: $max |Σ_{after} − I| \approx 10^{-7}$, and $mean |Σ_{after} − I| <10^{-7}$. This shows the success of whitening.
>
> **(2) Eigenvalues collapse from a wide spectrum (9–54k) to 1.0.** Before whitening, eigenvalues span several orders of magnitude; after whitening, all become exactly 1.0, confirming perfect decorrelation.
>
> | Layer | $\lambda_{\min}$ (Pre) | $\lambda_{\max}$ (Pre) | Mean (Pre) | Std (Pre) | After Deviation $\epsilon$ ($1-\lambda$)Max / Mean / Std |
> |:---:|:---:|:---:|:---:|:---:|:---:|
> | **3** | 22.34 | $3.92\times10^4$ | 571 | 726 | $1.0\text{e-}7$ / $5.0\text{e-}8$ / $1.7\text{e-}8$ |
> | **5** | 30.59 | $3.80\times10^4$ | 512 | 699 | $3.0\text{e-}8$ / $2.0\text{e-}8$ / $1.2\text{e-}8$ |
> | **7** | 17.59 | $4.32\times10^4$ | 534 | 814 | $6.0\text{e-}8$ / $3.0\text{e-}8$ / $2.1\text{e-}8$ |
> | **9** | 9.36 | $5.40\times10^4$ | 528 | 900 | $1.1\text{e-}7$ / $4.0\text{e-}8$ / $2.9\text{e-}8$ |
> | **12**| 9.60 | $5.25\times10^4$ | 577 | 929 | $1.0\text{e-}7$ / $4.0\text{e-}8$ / $2.6\text{e-}8$ |
> | **24**| 24.02 | $1.86\times10^4$ | 639 | 775 | $4.0\text{e-}8$ / $2.0\text{e-}8$ / $1.4\text{e-}8$ |
>
> **(3) Off-diagonal correlations are fully removed after Whitening.** Prior to whitening, gate covariance shows non-trivial correlations (up to 0.78). All are eliminated after whitening (Σ_after = I).
> | Layer | Mean | Std | Max |
> | :---: | :---: | :---: | :---: |
> | **3** | $2.68 \times 10^{-7}$ | $1.06 \times 10^{-2}$ | 0.72 |
> | **5** | $-1.07 \times 10^{-6}$ | $1.14 \times 10^{-2}$ | 0.64 |
> | **7** | $4.31 \times 10^{-7}$ | $1.27 \times 10^{-2}$ | 0.78 |
> | **9** | $4.02 \times 10^{-6}$ | $1.42 \times 10^{-2}$ | 0.63 |
> | **12**| $-5.05 \times 10^{-7}$ | $1.34 \times 10^{-2}$ | 0.76 |
> | **24**| $1.05 \times 10^{-6}$ | $1.01 \times 10^{-2}$ | 0.79 |
>
> All, we will include heatmaps, eigenvalue plots, and correlation histograms based on the above statistics in the appendix to further illustrate these findings.

---

> ### Author Response · Authors · 2025-11-21
> **Response to Reviewer hRUS- Part #6**
>
> ### Q4 Expand experimental scope
> >**Q4-Q1**: Evaluate on larger-scale models or expert number>16.
>
> **A**: We evaluate on two extra models for effectiveness verification,*Qwen2-57B-A14B (**64 experts**)* and Phi-3.5-MoE (16 experts). **Our method achieves lossless on 20% compression.**
>
> | Ratio | Method          | Layer Selected | Retain Experts | Layer Ratio | Wiki2   | PTB | C4 | Avg  | OBQA  | A_E  | WinoG  | A_C  | PIQA |  Avg  |
> |-------|-----------------|----------------|----------------|--------------|---------------|--------|--------|-------------|----------|------------|------------|------------|-----------|----------------|
> | origin | Uncompressed  | – | – | – | 5.87 | 10.87 | 9.14 | 8.63 | 0.33 | 0.77 | 0.74 | 0.48 | 0.81 | **0.626** |
> | 0.2 | MoE-SVD         | * | * | * | 5.41 | 13.26 | 11.63 | 10.10 | 0.30 | 0.74 | 0.73 | 0.45 | 0.78 | 0.600 |
> | 0.2 | DT-MoE          | 6  | 8  | 0.20 | 6.98 | 10.70 | 10.73 | 9.47 | 0.30 | 0.78 | 0.72 | 0.47 | 0.79 | 0.612 | 46.62 |
> | 0.2 | DT-MoE  | 8  | 16 | 0.20 | 6.75 | 10.60 | 10.85 | **9.4** | 0.33 | 0.79 | 0.72 | 0.49 | 0.80 | **0.626** |
> | 0.4 | NAEE            | * | * | * | 6.81 | 11.34 | 11.57 | 9.91 | 0.31 | 0.73 | 0.73 | 0.46 | 0.76 | 0.598 |
> | 0.4 | MoE-I2          | * | * | * | 24.90 | 77.05 | 22.50 | 41.48 | 0.26 | 0.70 | 0.46 | 0.41 | 0.75 | 0.516 |
> | 0.4 | DT-MoE  | 16 | 16 | 0.20 | 8.53 | 13.73 | 14.12 | 12.13 | 0.30 | 0.78 | 0.72 | 0.48 | 0.79 | **0.614** |
> | 0.4 | DT-MoE  | 14 | 8  | 0.40 | 8.01 | 12.39 | 12.90 | 11.10 | 0.30 | 0.77 | 0.71 | 0.45 | 0.78 | 0.602 |
> | 0.4 | DT-MoE  | 14 | 8  | 0.60 | 8.16 | 12.59 | 13.13 | 11.29 | 0.29 | 0.76 | 0.71 | 0.46 | 0.78 | 0.600 |
> | 0.6 | DT-MoE | 20 | 8  | 0.60 | 12.40 | 19.88 | 20.96 | 17.75 | 0.27 | 0.73 | 0.68 | 0.41 | 0.75 | **0.568** |
>
>
> *Qwen2-57B-A14B.*
> Specifically, we excluded the shared experts and tensorized the remaining experts. We provide different configurations to compose the target compression budget, such as fewer experts or higher intra-layer compression.
>
> | Ratio | Method   | W2  | PTB  | C4  | Avg | OBQA | A_e | Wino | Hella | A_c | PIQA | MathQA | Avg |
> |-------|----------|-----|------|-----|--------|------|--------|--------|--------|--------|-------|--------|------|
> | origin | baseline | **3.5** | **8.4** | **8.2** | **6.7** | .40 | .77 | .76 | .68 | .56 | .79 | .38 | .62 |
> | 0.2 | ASVD      | 7.2 | 10.7 | 9.6 | 9.2 | .35 | .73 | .72 | .57 | .49 | .75 | .34 | .56 |
> |     | SVD-LLM   | 8.3 | 14.8 | 12.9 | 12.0 | .31 | .67 | .66 | .53 | .45 | .72 | .22 | .51 |
> |     | MoE-SVD   | **4.6** | 10.1 | 9.9 | 8.2 | .39 | .77 | .73 | .63 | .53 | .78 | .35 | .60 |
> |     | **Ours**  | 4.7 | **9.2** | **9.1** | **7.7** | **.39** | **.77** | **.74** | **.65** | **.55** | **.79** | **.38** | **.61** |
> | 0.4 | NAEE      | 8.2 | 20.1 | 16.1 | 14.8 | .35 | .73 | .73 | .61 | .48 | .76 | .37 | .57 |
> |     | MoE-I2    | 7.5 | 21.0 | 21.0 | 16.5 | .29 | .59 | .67 | .27 | .40 | .70 | .25 | .45 |
> |     | SVD-LLM   | 38.8 | 68.5 | 43.8 | 50.4 | .23 | .56 | .60 | .39 | .31 | .66 | .24 | .43 |
> |     | MoE-SVD   | **5.5** | 11.7 | 11.9 | 9.7 | .35 | .72 | .72 | .58 | .48 | .75 | .31 | .56 |
> |     | **Ours**  | 6.4 | **10.9** | **10.8** | **9.4** | **.35** | **.75** | **.73** | **.61** | **.50** | **.78** | **.33** | **.58** |
> | 0.6 | ASVD      | 107.7 | 208 | 161 | 159 | .18 | .40 | .53 | .30 | .24 | .59 | .23 | .35 |
> |     | SVD-LLM   | 7168 | 7101 | 7119 | 7129 | .15 | .28 | .51 | .26 | .22 | .54 | .21 | .31 |
> |     | MoE-SVD   | **7.5** | 21.0 | 21.9 | **16.8** | .30 | .60 | .68 | .46 | .40 | .71 | **.25** | .49 |
> |     | **Ours**  | 13.7 | **19.7** | **17.9** | 17.1 | **.28** | **.70** | **.68** | **.46** | **.41** | **.73** | .23 | **.50** |
>
> *Phi-3.5-MoE with 16 experts.*
> For Phi-3.5-MoE (16 experts), our approach consistently outperforms existing baselines under medium expert number MoE models.
> >**Q4-Q2**: Calibration set size sensitivity analysis.
>
> **A**: We vary the calibration set size (128–2048) and also compare different calibration corpora (WikiText-2 vs. PTB). The results show that calibration size has a negligible impact once $𝑁>128$, and the method is robust to the choice of calibration corpus.
>
> | Calib. Set | Wiki2 | PTB | C4 | **Avg**  | OBQA | ARC-E | Wino | Hella | ARC-C | PIQA | Math | **Acc**  |
> | :--- | :---: | :---: | :---: | :---: | :---: | :---: | :---: | :---: | :---: | :---: | :---: | :---: |
> | Wiki-128    | 4.50 | 17.3 | 7.73 | 9.84 | .33 | .83 | .77 | .64 | .53 | .81 | .40 | .62 |
> | Wiki-256    | 4.49 | 17.2 | 7.73 | 9.81 | .33 | .83 | .77 | .64 | .53 | .82 | .40 | .62 |
> | Wiki-512    | 4.49 | 17.2 | 7.73 | 9.82 | .33 | .83 | .76 | .64 | .53 | .81 | .40 | .61 |
> | Wiki-1K     | 4.49 | 17.3 | 7.72 | 9.83 | .33 | .83 | .77 | .64 | .54 | .81 | .40 | .62 |
> | Wiki-2K     | 4.49 | 17.3 | 7.72 | 9.82 | .33 | .83 | .76 | .64 | .53 | .81 | .40 | .61 |
> | PTB-256     | 4.51 | 17.2 | 7.74 | 9.83 | .32 | .82 | .77 | .64 | .52 | .81 | .40 | .61 |

---

> ### Author Response · Authors · 2025-11-21
> **Response to Reviewer hRUS- Part #7**
>
> >**Q4-Q3**: Broaden task diversity: add code generation (HumanEval), multilingual tasks (XNLI/XQuAD)
>
> **A:** We thank the reviewer for this suggestion. We have added HumanEval (code generation) and XNLI (multilingual NLI) tasks for further evaluation. We did not additionally include scientific QA task because our current evaluation tasks (e.g. ARC-C, ARC-E) already covers harder scientific reasoning tasks. Below we report the results against MoE-SVD under 0.2 and 0.4 compression budgets using Mixtral-8x7B.
>
> | Task        | Ratio | MoE-SVD | TD-MoE | Δ  |
> |-------------|:-----:|:--------:|:-------:|:----------------------:|
> | HumanEval (Pass@1) | origin | — |0.293 | — |
> | | 0.2 | 0.079 | 0.240 | **+0.161** |
> | | 0.4 | 0.037 | 0.140 | **+0.103** |
> | XNLI (Acc.) | origin | — | 0.450 | — |
> | | 0.2 | 0.390 | 0.440 | **+0.050** |
> | | 0.4 | 0.380 | 0.410 | **+0.030** |
>
> Our method shows higher performance on both tasks. MoE-SVD collapes on HumanEval, the sensitive code task. This is because that MoE-SVD truncates each expert independently and destroys shared code-specific subspaces, and HumanEval is highly sensitive to such lost structure, so small decomposition errors cause the Pass@1 score to collapse.

---

> > ### Comment · Reviewer_hRUS · 2025-11-25
> >
> > Thank you. I had updated my rating.

---

> > > ### Author Response · Authors · 2025-11-25
> > >
> > > We authors thank you for the update and for kindly revisiting your evaluation.
> > > We appreciate your thoughtful feedback during the rebuttal process and are glad the clarifications were helpful.

---

### Comment · Area_Chair_n3LE · 2025-11-26
**Reviewer & Author Discussion**

Dear Reviewers,

We kindly encourage you to review and respond to the authors’ rebuttals. Your timely feedback is important for ensuring a fair and thorough review process. Thank you for your contributions to ICLR 2026.

Thank you very much for your time and support.

Best regards,

Area Chair n3LE

---

### Meta-Review · Area_Chair_gNZe · 2026-01-03

**Summary:**

This paper studies model compression in the context of mixture-of-expert models, proposing an approach based on tensor decomposition that aims to capture shared redundancies across experts in a straightforward manner. The most salient concerns raised by reviewers included the limited number of models evaluated, lack of analysis of combinations with other compression techniques, and a missing discussion of compute costs.

**Reviewer Concerns:**

The authors partially addressed the model coverage criticism, evaluating on larger models of the same class but not different models beyond Qwen and Phi. The authors addressed the combination with other compression techniques to a significant extent, reporting evaluations with quantization and pruning. The authors fully addressed the question of computational cost.

**Reviewer Scores:**

Author and reviewer comments indicate that Reviewer hRUS increased their score from 6 to 8.
I imagine Reviewer E6zg would have increased their score to a 6.
Reviewer oj4s already gave the largest possible score and would have kept it at 10.

---

### Decision · Program_Chairs · 2026-01-26

Accept (Poster)